# Comparison of FORCE trained spiking and rate neural networks shows spiking networks learn slowly with noisy, cross-trial firing rates

**Thomas Robert Newton** [iD] [1,2], **Wilten Nicola** [iD] [1,2,3]*

**1** Department of Mathematics and Statistics, University of Calgary, Calgary, Canada, **2** Hotchkiss Brain Institute, University of Calgary, Calgary, Canada, **3** Department of Cell Biology and Anatomy, University of Calgary, Calgary, Canada

* wilten.nicola@ucalgary.ca

**Data availability statement:** All code is available at
https://github.com/Trnewton/RateVsSpike [44].

## Abstract

Training spiking recurrent neural networks (SRNNs) presents significant challenges compared to standard recurrent neural networks (RNNs) that model neural firing rates more directly. Here, we investigate the origins of these difficulties by training networks of spiking neurons and their parameter-matched instantaneous rate-based RNNs on supervised learning tasks. We applied FORCE training to leaky integrate-and-fire spiking networks and their matched rate-based counterparts across various dynamical tasks, keeping the FORCE hyperparameters identical. We found that at slow learning rates, spiking and rate networks behaved similarly: FORCE training identified highly correlated weight matrix solutions, and both network types exhibited overlapping hyperparameter regions for successful convergence. Remarkably, these weight solutions were largely interchangeable—weights trained in the spiking network could be transferred to the rate network and vice versa while preserving correct dynamical decoding. However, at fast learning rates, the correlation between learned solutions dropped sharply, and the solutions were no longer fully interchangeable. Despite this, rate networks still functioned well when their weight matrices were replaced with those learned from spiking networks. Additionally, the two network types exhibited distinct behaviours across different sizes: faster learning improved performance in rate networks but had little effect in spiking networks, aside from increasing instability. Through analytic derivation, we further show that slower learning rates in FORCE effectively act as a low-pass filter on the principal components of the neural bases, selectively stabilizing the dominant correlated components across spiking and rate networks. Our results indicate that some of the difficulties in training spiking networks stem from the inherent spike-time variability in spiking systems—variability that is not present in rate networks. These challenges can be mitigated in FORCE training by selecting appropriately slow learning rates. Moreover, our findings suggest that the decoding solutions learned by FORCE for spiking networks approximate a cross-trial firing rate-based decoding.

**Funding:** This work was funded by an NSERC discovery grant (DGECR/00334-2020 to WN) and a Tier II Canada Research Chair in Computational Neuroscience (CRC-2019-00416 to WN). The funders had no role in study design, data collection and analysis, decision to publish, or preparation of the manuscript.

**Competing interests:** The authors have declared that no competing interests exist.

## Author summary

Training spiking neural networks is much harder compared to training standard recurrent neural networks that are more closely tied to neural firing rates. To understand why, we trained parameter matched spiking and rate-based networks on the same supervised learning tasks with the FORCE technique. We found that the learned spike weights were highly correlated and interchangeable across spiking and firing rate networks for slow learning rates. However, when both networks learn fast, the spiking networks show no tangible improvements in their performance in comparison to the rate networks, with instabilities caused by faster learning in the spiking network. These networks also discover uncorrelated solutions to their weights when the learning is fast, that are only interchangeable in one direction, from spike to rate. This suggests that the decoding solutions learned by FORCE for spiking networks approximate a cross-trial firing rate-based decoding. We then analytically determine that, with slower learning rates, FORCE acts as a low-pass filter on the principal components of the neural bases, where the leading components are highly correlated across spiking and rate networks.

## Introduction

One way that neurons communicate with one-another is through action potentials or spikes [1,2]. A central question in neuroscience is how neurons use these spikes to encode information and perform computation. One perspective is that individual spikes are not themselves meaningful; rather, the relevant quantity for computation is the firing rate or frequency of at which spikes are transmitted between neurons [3,4]. In contrast, it has been suggested that the precise or relative timing of spikes is important for neural computation [4]. It is sensible to consider which representation—rate or spike—captures the essential information for a given system and modelling task. We define a system as rate-encoded if a firing rate description captures most of the system's behaviour, making the precise timing of individual spikes negligible. Conversely, we describe a system as spike-encoded if a purely rate-based description fails to capture its behaviour and spike timing is essential. For a more in-depth discussion of the rate versus spike timing debate, see Brette (2015) [4]. Here, we demonstrate that in many cases, a cross-trial firing rate model is sufficient to capture the information utilized by the FORCE learning technique to training spiking neural networks.

   Learning low-dimensional dynamical systems is one such task where the difference between spikes and rates becomes readily apparent. There has been tremendous success in training recurrent neural networks with a variety of techniques to learn low-dimensional dynamical systems [5–9], however, learning in spiking networks is considerably more difficult. One of the most important differences between spiking and rate based neurons is the techniques used to train their resulting networks. Spike based neurons are typically more difficult to train and require specialized techniques due to discontinuities in the differential equations used to phenomenologically describe spiking (integrate-and-fire neurons) and synaptic propagation, which complicates the computation of derivatives. As a result, gradient based optimization routines cannot be immediately applied to spiking networks. In contrast, rate based neurons are readily trained with gradient based techniques and have demonstrated remarkable flexibility and performance [10]. Recently, there has been developed a range of different approaches to training spiking neural networks. For example the neural engineering framework (NEF) [11], predictive coding [12–15], surrogate gradient [16,17], and FORCE [6,7,18].

All of these training techniques use a top-down approach that uses a prescribed target task (supervisor) to determine the connectivity of the network required to perform the task. In the NEF, neuronal tuning curves are used to optimize for the weights while predictive coding uses fast, precise synaptic interactions between neurons to balance neural activity. Though, in the case of predictive coding, it has been shown that it is possible to use local biologically plausible learning rules to learn the required synaptic connections required to create this balanced activity [19,20]. Surrogate gradient methods differ from other techniques by using a smoothed approximation of the neuron's derivative with respect its input current, known as a pseudo-derivative, to enable gradient descent-based backpropagation for network training [21]. However, all three of these approaches depend on the specific types of neuron models used in the networks. In contrast, FORCE based techniques are agnostic to the underlying network model being used. This offers the advantage that it can be applied to train networks with both rate [6] and spike based neurons [18] without major modifications. Nonetheless, there are underlying differences in FORCE trained rate and spiking neural networks, which as of now have not be thoroughly investigated.

Here we compare FORCE trained leaky-integrate-and-fire (LIF) networks and their corresponding parameter-matched instantaneous firing rate networks. We first introduce the Leaky-integrate-and-fire (LIF) spiking model and an corresponding LIF instantaneous firing rate neuron model. The firing rate model computes the expected firing rate of a neuron based on its instantaneous input current, effectively capturing the time-averaged firing behaviour at each moment. Then, we connect these two types of neuron models into spiking networks and rate networks, which we demonstrate can both be trained using FORCE [6,18], with exact matches for the initial weights, neuronal parameters, and FORCE hyperparameters in both networks. Our goal in introducing the rate model is not to claim a general equivalence between spiking and rate networks, but rather to investigate how the FORCE algorithm leverages the information specific to each model. If, after training, the two networks exhibit similar behaviour, we can conclude that the distinction between the models is inconsequential in the context of FORCE training.

We found that when the learning rate is slow, the learned connectivity structure of the spiking and firing rate networks were highly correlated to the point that learned weight solutions could be interchanged, with the learned weights of the spiking network leading to successful task dynamics in the rate network, and vice-versa. However, substantial differences emerged when the learning rate was fast. First, the correlation between the learned weights dropped sharply and interchangeability became one-directional: weights from the spiking network could still function in the rate network, but not the other way around. Second, we found that the spiking network had inherently noisier neural outputs, which resulted in worse error scaling compared to the rate network, and prevented the learned connectivity from converging to a stationary structure. Third, rate networks benefit from faster learning rates, with reduced errors and larger areas of convergence in hyperparameter space, with no discernible improvements by the spiking networks. These findings suggest that some of the challenges associated with FORCE training in spiking neural networks stem from the algorithm learning a noisy, trial-averaged firing rate solution. This solution destabilizes at higher learning rates, contributing to the difficulties in training.

## Results

### Network models

To investigate the differences in spiking and rate based neural networks (Fig 1), we used the FORCE method to train both spiking neural networks and their corresponding parameter-matched instantaneous firing rate networks on a range of tasks [6,18]. The corresponding rate model is derived in Materials and Methods by analyzing the steady-state firing rate of an isolated LIF neuron under constant input current. Remarkably, when trained with FORCE, the resulting LIF and firing rate networks exhibit closely related properties. By using exactly parameter matched networks, we can compare the weight matrix solutions derived from spiking and rate networks to determine if these solutions are interchangeable and under what conditions. If the weights can indeed be interchanged, this implies that the networks are operating in functionally similar ways.

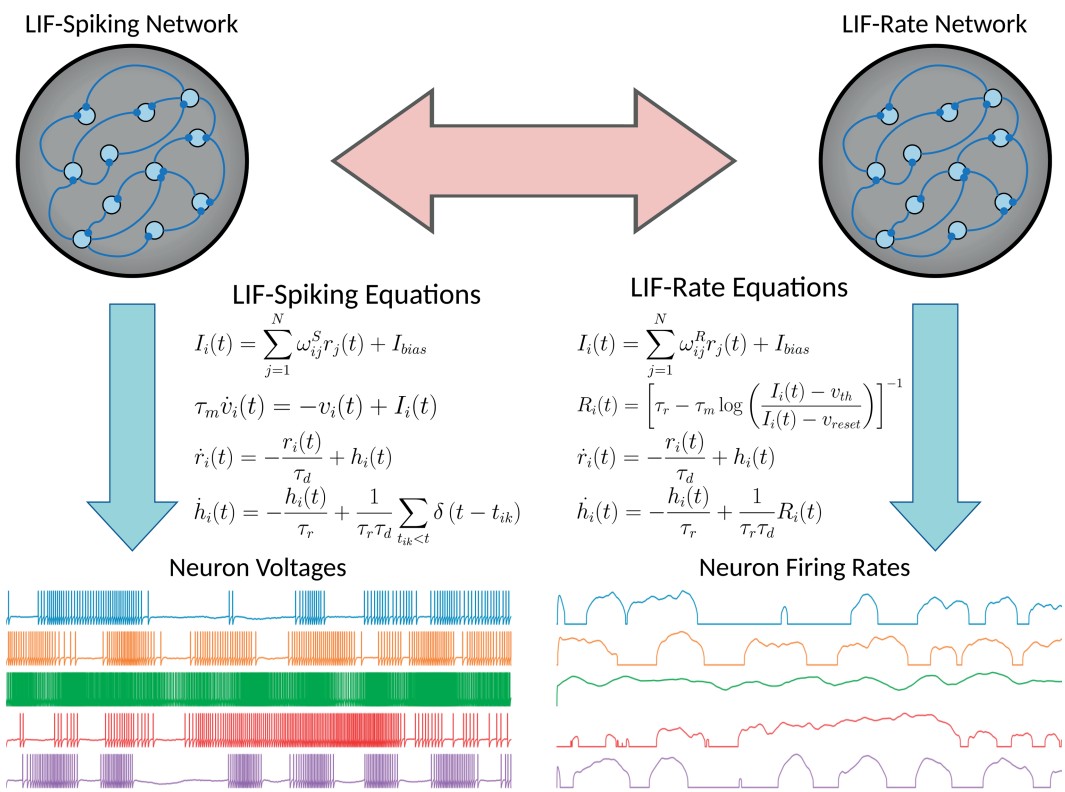

**Fig 1. Leaky Integrate-and-Fire spiking and Equivalent rate networks.** In standard spiking networks, each neuron has a membrane voltage governed by linear dynamics. Once this voltage crosses a threshold, a spike is fired by the neuron which is then filtered by a set of double exponential filter equations. The resulting filtered spike current or post-synaptic current is then used as the output from the neuron. In the firing rate network model, compute the instantaneous theoretical firing rate given the input current for each neuron. This firing rate is then again filtered by the double exponential filter equations to produce the post-synaptic current as the neurons output.

We considered networks of $N$ leaky integrate-and-fire neurons, with membrane time constants $\tau_m$, and double-exponential filtered synapses:

$$\tau_m \dot{v}_i(t) = -v_i(t) + I_i^S(t) \tag{1}$$

$$\dot{h}_i^S(t) = -\frac{h_i^S(t)}{\tau_r} + \frac{1}{\tau_r \tau_d} \sum_{t_{ik} < t} \delta(t - t_{ik}) \tag{2}$$

$$\dot{r}_i^S(t) = -\frac{r_i^S(t)}{\tau_d} + h_i^S(t) \tag{3}$$

$$\hat{x}^S(t) = \sum_{i=1}^{N} \phi_i^S r_i^S(t) \tag{4}$$

where $\tau_r$ and $\tau_d$ are the synaptic rise and decay times. We also note that we have absorbed a unit resistance into the input current, so that $I_i^S(t)$ in (1) can be expressed in volts. See Leaky Integrate-and-Fire Network section of the Material for more details.

The voltage of the $i$th neuron is given by $v_i(t)$. Neuron $i$ fires its $k$th spike at time $t_{ik}$ when the threshold voltage is crossed $v_i(t) = v_{th}$. After each spike, the voltage $v_i(t)$ is reset to $v_{reset}$ and held there for a fixed refractory period ($\tau_{ref}$). Each neuron receives a current, given by:

$$I_i^S(t) = \sum_{j=1}^{N} \omega_{ij}^S r_j^S(t) + I_{bias} \tag{5}$$

where $r_j^S(t)$ is the postsynaptic current given in (3) due to the filtered spikes and $I_{bias}$ is a background bias current which was set at the rheobase ($I_{bias} = -40$ mV). The connectivity weight matrix $\omega_{ij}^S$ is trained by the FORCE algorithm so that the spiking network can approximate some $M$-dimensional dynamical system, $x^S(t)$. The approximant for this system is given by $\hat{x}^S(t)$ which is computed using a linear decoder $\phi^S$ applied to the filtered spikes, $r^S(t)$. The neuronal parameters used across both network types are listed in Table 1 and explained in detail in the Leaky Integrate-and-Fire Network section of the Material.

In FORCE trained networks (Fig 2), the synaptic connectivity matrix $\omega^S \in \mathbb{R}^{N \times N}$ consists of the sum of random static weights and a learned, low rank perturbation to the weight matrix:

$$\omega^S = G\omega^0 + Q\eta \left[ \phi^S \right]^T \tag{6}$$

where $\omega^0 \in \mathbb{R}^{N \times N}$ is the reservoir weight matrix, $\eta \in \mathbb{R}^{N \times M}$ is the feedback encoder matrix, and $\phi^S \in \mathbb{R}^{N \times M}$ the linear decoder weight matrix for the spiking network, with $M$ being the dimension of the target dynamics $x^S(t)$. The static portion of the network weights $G\omega^0$ consists of a scalar strength parameter $G$ and a sparse random matrix $\omega^0$. The static weights serve to initialize the network into a regime of high-dimensional chaotic dynamics, which we control with the parameter $G$. The other portion of the weight matrix consists of a learned decoder $\phi^S$, a static random encoder $\eta$, and a scalar strength parameter $Q$. The learned linear decoder $\phi^S$ serves the primary task of decoding the neural dynamics to produce the network

**Table 1. Leaky Integrate-and-fire neuron model parameters.**

| Parameter | $\tau_r$ | $\tau_d$ | $\tau_m$ | $\tau_{ref}$ | $v_{reset}$ | $v_{th}$ | $I_{bias}$ |
|---|---|---|---|---|---|---|---|
| Value | 2 ms | 20 ms | 10 ms | 2 ms | -65 mV | -40 mV | -40 mV |

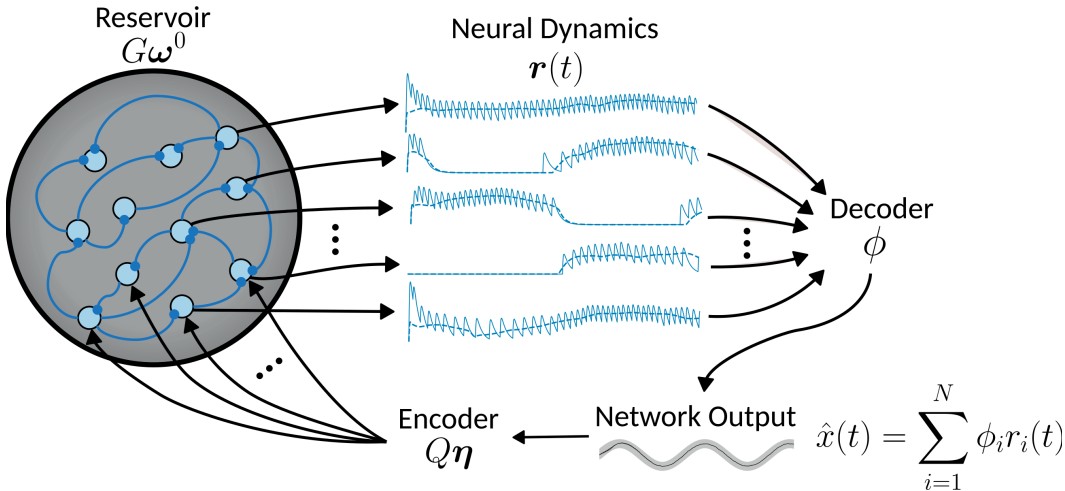

**Fig 2. FORCE training spiking LIF and LIF-matched rate networks.** The single layer recurrent neural network used in the FORCE method consists of: a set of fixed reservoir weights, a set of fixed encoder weights and a set of learned decoder weights. The reservoir network creates a chaotic pool of rich mixed dynamics which are used to linearly decode the target supervisor $\boldsymbol{x}(t)$ with $\hat{\boldsymbol{x}}(t)$. This decoder $\boldsymbol{\phi}_S/\boldsymbol{\phi}_R$ (S for spikes, R for rates) is learned online using the Recursive Least Squares (RLS) algorithm. The encoder weights $\boldsymbol{\eta}$ then feedback the decoded output $\hat{x}(t)$ into the reservoir to stabilize the dynamics.

output $\hat{\boldsymbol{x}}^S(t)$, but also works in conjunction with the encoder to stabilize the network dynamics via the feedback connections. The encoder $\eta$ partially defines the tuning properties of the neurons to the learned output dynamics. The parameter $Q$ is used to control the strength of the feedback to balance against the chaotic dynamics induced by static weight matrix $G\omega^0$ controlled by the parameter $G$. By introducing the $Q$ and $G$, parameters the macro-scale dynamics of the network can be controlled; either making the system more (less) chaotic by increasing (decreasing) $G$ or creating more (less) structured heterogeneity in the neural firing rates by increasing (decreasing) $Q$ [22,23].

The random matrix $\omega^0$ was generated from a normal distribution with mean 0 and variance of $\left(Np^2\right)^{-1}$ where $p$ is the sparsity of the matrix and $N$ the number of neurons. This scales the weights to be in proportion to the inverse of the square-root of the number of connections [18,23]. We additionally considered the case where the sample mean of each row in the matrix was explicitly set to be 0 [18]. The components of the encoder $\eta$ are drawn from a uniform distribution over the $M$ dimensional unit square $[-1, 1]^M$, where $M$ is the dimension of the target dynamics.

The LIF network above has a corresponding system of instantaneous rate equations [1,24]:

$$R_i(t) = \left[\tau_r - \tau_m \log\left(\frac{I_i^R(t) - v_{th}}{I_i^R(t) - v_{reset}}\right)\right]^{-1}$$

$$\dot{h}_i^R(t) = -\frac{h_i^R(t)}{\tau_r} + \frac{1}{\tau_r\tau_d}R_i(t)$$

$$\dot{r}_i^R(t) = -\frac{r_i^R(t)}{\tau_d} + h_i^R(t)$$

$$\hat{\boldsymbol{x}}^R(t) = \sum_{i=1}^{N} \phi_i^R r_i^R(t)$$

where $R_i(t)$ acts as the firing rate for neuron $i$ as a function of input current $I_i^R(t)$:

$$I_i^R(t) = \sum_{j=1}^{N} \omega_{ij}^R r_j^R(t) + I_{bias}$$

$$\omega^R = G\omega^0 + Q\eta \left[\phi^R\right]^T$$

where we have introduced the new linear decoder matrix $\phi^R \in \mathbb{R}^{N \times M}$ for the firing rate network. The firing rate referred to here should be understood as the instantaneous firing rate averaged over time for a single neuron, as opposed to being averaged over a population of neurons or across multiple trials [4].

To determine how both networks learn low dimensional dynamics, the parameters for both the instantaneous rate network and the LIF spiking network were exactly matched, down to the initial static weight matrix $G\omega^0$ and randomly generated encoders $\eta$. After training, each model will have an associated decoder $\phi^S$ ($S$ for Spikes) and $\phi^R$ ($R$ for Rates). Thus, to compare and contrast the differences and similarities between spiking and rate networks, we can compare the decoders $\phi^R$ and $\phi^S$, as all other parameters are matched (Fig 1). Additionally, we can compare the set of temporal features, or neural basis, generated by each network, which is represented by the filtered post-synaptic currents $r^S(t)$ and $r^R(t)$.

The FORCE technique was previously used to train rate-based hyperbolic tangent networks [6] and a variety of spiking neural networks [18]. FORCE training applies online updates to the learned decoder $\phi$ using the Recursive Least Squares (RLS) supervised learning method [6,25] with updates:

$$e(t) = \hat{x}(t) - x(t)$$
$$\phi(t) = \phi(t - \Delta t) - e^T(t)P(t)r(t)$$
$$P(t) = P(t - \Delta t) - \frac{P(t - \Delta t)r(t)r^T(t)P(t - \Delta t)}{1 + r^T(t)P(t - \Delta t)r(t)}$$

and initialization:

$$\phi(0) = \mathbf{0}_{N \times M}$$
$$P(0) = \lambda^{-1}I_N = \alpha I_N$$

where $\lambda$ is the regularization parameter or inverse of the learning rate $\alpha$, and $\mathbf{0}_{N \times M}$ is the zero matrix. RLS minimizes the mean squared error between the target dynamics $x(t)$ and the network's output dynamics $\hat{x}(t)$. To prevent large weights (decoders), RLS incorporates Tikhonov regularization [26–28], controlled by the ridge parameter $\lambda$. The ridge parameter influences both the regularization and is the inverse of the learning rate $\alpha$. Initially, networks are run with their decoder set to zero, allowing the generation of rich neural dynamics before RLS is activated to update the decoder. Training is considered successful if the network can continue to display the target dynamics without supervision (RLS Off). For more details see FORCE Training section in Materials and Methods.

## FORCE can train LIF and parameter matched rate networks

To demonstrate that spiking and LIF-matched rate networks are comparable when trained using the FORCE method, we trained several networks to learn various dynamical systems tasks and compared the resulting neural dynamics across spiking and rate networks

(Fig 3A). The tasks included a range of oscillator problems with varying dimensionalities and complexities, a random kick Pitchfork input-output task, and the chaotic Lorenz system. We focused on three primary types of oscillator supervisors: a simple sinusoid, a higher-dimensional Fourier supervisor, and the Ode to Joy supervisor, which features long periodicity. The Fourier supervisor is an $n$-dimensional supervisor consisting of $n$ sinusoidal waves with frequencies $1, 2, ... n$. The Ode to Joy supervisor comprises six components, with pulses spaced to mimic the notes of the song 'Ode to Joy.' In some cases we additionally add a high-dimensional temporal signal (HDTS) to the Ode to Joy supervisor. The HDTS consists of a series of pulses in additional components that acts as a clock for longer signals [18]. The Pitchfork task consists of a pitchfork normal form, which has two stable fixed points, which receives random kicks as inputs that can cause the supervisor to change fixed point. For further details of the supervisors, refer to the Materials and Methods section.

During the initial spontaneous activity phase, both network types exhibit irregular asynchronous activity in their dynamics, reflecting the chaotic state. However, during learning, the neural dynamics rapidly converge and become highly correlated between the two network types (Fig 3A). This convergence coincides with rapid adjustments to the decoders, causing the decoded output to quickly match the target supervisor. The magnitude of these rapid updates to the decoders is also highly correlated between the spike and rate networks (Fig 3A). Once the decoders and dynamics have stabilized, the RLS learning algorithm can be turned off. If the learning process was successful and the target dynamics are not chaotic, the resulting learned network dynamics show a high degree of correlation between the rate and spiking networks (Fig 3A). This pattern holds true for various periodic supervisors, including more complex oscillators (Fig 3C) and higher-dimensional periodic signals, such as those in "Ode to Joy" (Fig 3D) [18]. In general, we found that as long as the spiking and rate networks had a shared driving input current, they would demonstrate highly correlated neural basis elements, and even showed correlated readout signals for randomly initialized decoders (S3 Fig).

The correlation between the rate and spiking network dynamics trained with FORCE is not limited to periodic functions. We also demonstrated that both types of networks can learn dynamical systems with inputs, such as the pitchfork system (Fig 3B) and chaotic systems like the Lorenz system (Fig 3F–3G). For the pitchfork system, there was a high degree of correlation between the two network dynamics both during and after training. In contrast, with the Lorenz system, we observed a high degree of correlation during training, but the neural dynamics of the two networks diverged after RLS was turned off as seen by differences in the neural currents. This divergence is due to the chaotic nature of the Lorenz system. Despite this divergence, the networks continued to produce qualitatively similar neural dynamics. The output dynamics also qualitatively matched across networks and with the target supervisor, with both networks producing tent maps consistent with those expected from the target Lorenz system [29] (Fig 3F–3G).

## Learned weights are interchangeable between LIF and parameter matched rate networks for slow learning rates

After demonstrating that both spiking LIF and LIF-matched rate networks could learn similar tasks and exhibit similar neural dynamics, we investigated under what parameter regimes did the correlation across networks hold, and whether correlated dynamics were reflected in the correlation between their decoders $\phi^S$ and $\phi^R$? In particular, is it possible to use the same learned weights for both the spiking network and the corresponding firing rate network? We trained each network type on a range of tasks, sweeping the $(Q, G, \alpha)$ parameter space, and

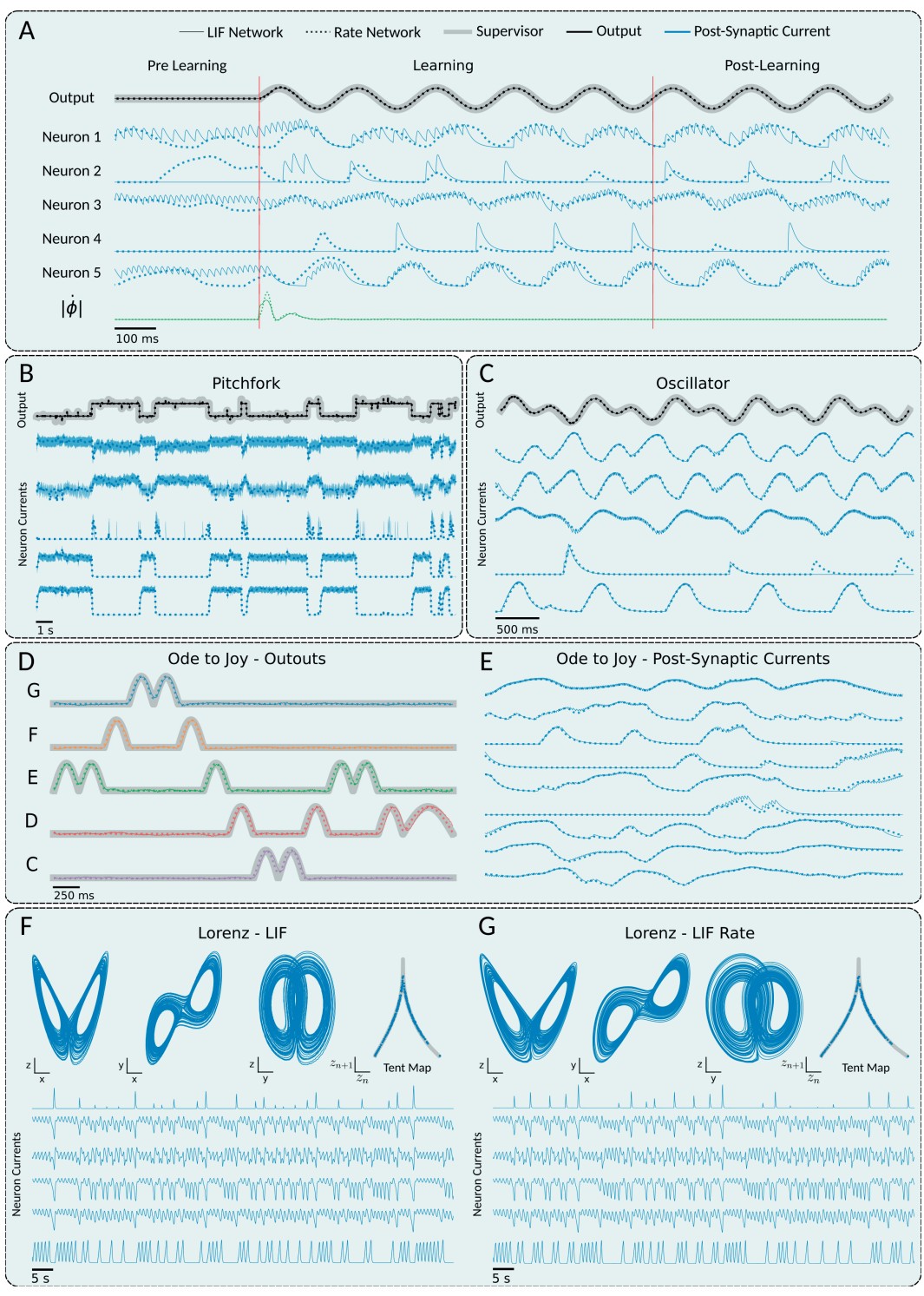

**Fig 3. FORCE can train both spiking LIF and LIF-matched rate networks.** Spiking LIF and parameter matched LIF-matched rate networks are trained on different tasks. The network outputs and sample neuron firing rates are overlaid. Spiking LIF values are plotted with a solid line, LIF-matched rate values are plotted with a dotted line, and supervisors with a thick grey line. **A** FORCE training can be broken down into three phases: pre-learning, learning, and post-learning. Before learning, the network dynamics are spontaneously chaotic. During learning, the network output is forced to match the target output, and the network dynamics are stabilized accordingly. After learning, if the training is successful, the network output and dynamics will continue to reproduce those stabilized during learning without any further weight updates.

The green line indicates the change in the Euclidean norm of the decoder. Across all three stages of learning, the neural dynamics and network outputs of the spiking and rate networks are highly correlated. **B** Networks of 2000 neurons were trained to reproduce the random kick pitchfork system using 120s of training, with 27s of testing displayed. **C** Networks of 2000 neurons were chaotically initialized, then trained to reproduce the product of a 1Hz and 2Hz sine wave using 5s of training, with 5s of testing displayed. **D–E** Networks of 2000 neurons were chaotically initialized, then trained to reproduce the first bar of the song "Ode to Joy" by Beethoven. Each of the 5 notes in the first bar was converted to a component of a 5-dimensional target signal. Quarter notes were represented by the positive portion of a 2Hz sine wave, and half notes by the positive portion of a 1Hz sine wave. Training consisted of 80s or 20 bar repetitions, while the testing displayed consists of 4s or 1 repetition. **F–G** Networks of 5000 neurons were randomly initialized into chaos, then trained to reproduce the global dynamics of the Lorenz system with parameters $\rho = 28$, $\sigma = 10$, and $\beta = \frac{8}{3}$. To train the networks, 200s of Lorenz target trajectory was used, and then 200s of testing output is displayed. Each of the 3 components of the Lorenz system was used to train a component of the 3-dimensional network output.

compared the correlation of the decoders across the network types. The $G$ and $Q$ parameters were taken in the ranges [0,0.2] and [2,30], which were chosen so as to initialize the networks into chaotic regimes that were still learnable (S1 Fig).

We observed that the correlations between decoders relied on both networks having successfully learned the task and was contingent on the learning rate $\alpha$. Specifically, for a slow learning rate (equivalently, high regularization) of approximately $\alpha \sim 10^{-5}$ or slower, we found a high correlation between the rate network's learned decoder $\phi^R$ and the spiking network's learned decoder $\phi^S$ (see Fig 4). This correlation was sufficiently strong that, in many cases, the decoders could be interchanged across network types and still have the networks retain the ability to reproduce the target dynamics without requiring further training. Since we also found that, in the case of a stabilizing input current, the bases generated by the spiking and the rate networks were highly correlated (S3 Fig). This implies that when the spiking network dynamics is being stabilized by an input current, as is the case with the feedback current in FORCE, the firing rate model effectively describes the spiking networks behaviour.

The correlation was also robust to different supervisors (Fig 4A–4C) such as the pitchfork, Ode to Joy, and sinusoidal signals with slow learning rates for both networks across a grid of $(Q,G)$ values. Subsequently, we selected the $(Q,G)$ point that yielded the highest Pearson correlation coefficient $\rho_{\phi^R,\phi^S}$ between the learned decoders $\phi^R$ and $\phi^S$ (Fig 4D–4F). We found a high degree of correlation across the learned neural dynamics, as well as a strong correlation across the decoders ($\rho = 0.61, 0.87$ and $0.74$ for pitchfork, Ode to Joy, and 5 Hz sine supervisors, respectively). In addition, we considered faster supervisors and training with very short training times (S2 Fig).

Further, we exchanged the decoders between the network types, running the spiking network with the firing rate decoder $\phi^R$ and the rate network with the spiking decoder $\phi^S$ (Fig 4G–4I). The resulting output dynamics and neural dynamics for the networks with swapped decoders were qualitatively similar to those of the original trained networks. Additionally, we observed a slight time dilation or contraction in the output dynamics after swapping decoders, stemming from subtle differences in the time scales of the firing rate approximation.

When the learning rate was fast ($\alpha > 10^{-4}$), we observed a diminished correlation between $\phi^R$ and $\phi^S$ (Fig 5). Nevertheless, we noted that the area of convergence in the $(Q,G)$ hyperparameter space tended to be larger and achieving lower testing errors for the rate network. Across a grid of $(Q,G)$ values, we simulated the pitchfork, Ode to Joy, and sinusoidal signals for both networks and computed the Pearson correlation coefficient $\rho_{\phi^R,\phi^S}$ across the learned decoders (Fig 5A–5C). Subsequently, we plotted the learned network outputs and neural

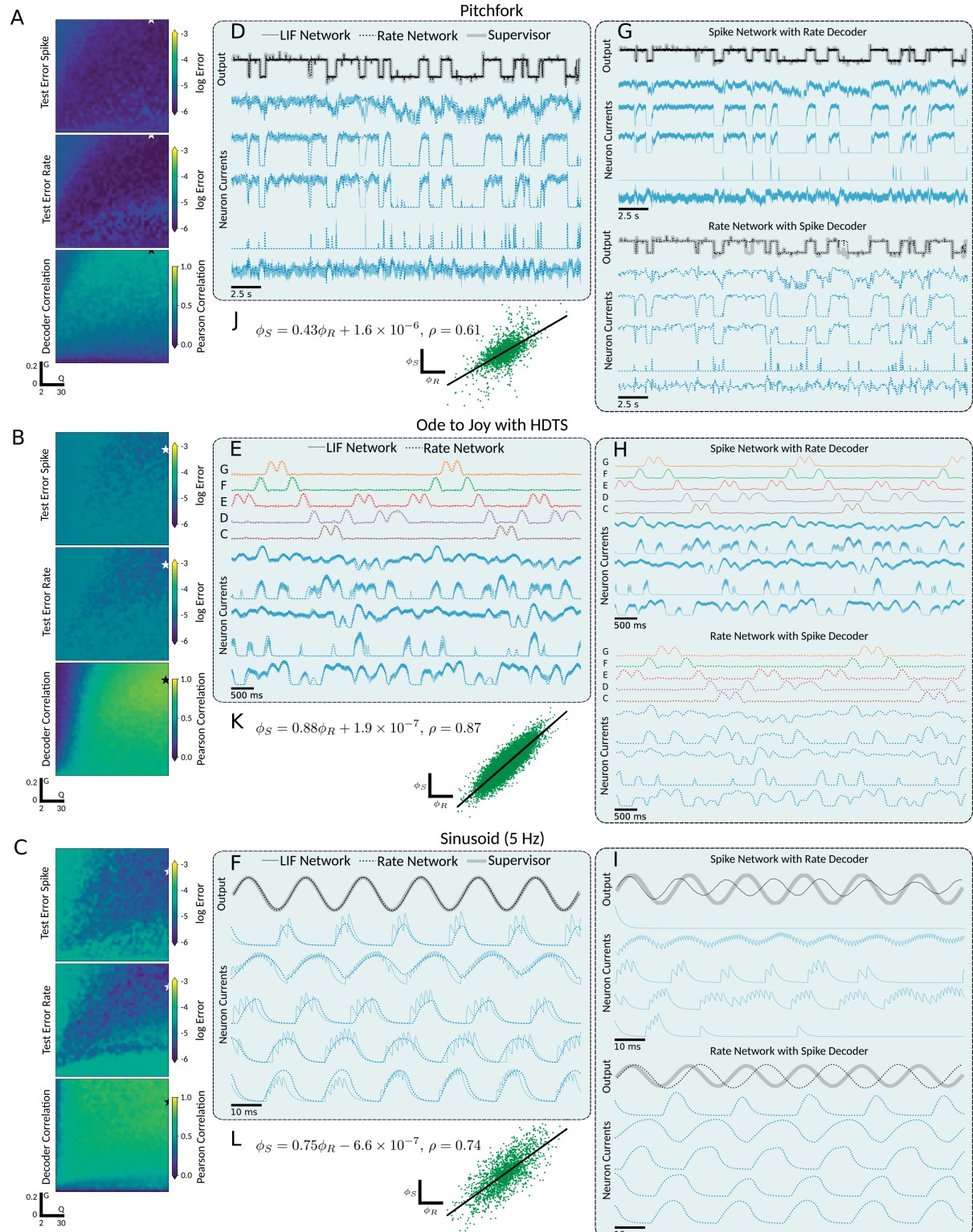

**Fig 4. FORCE training with slow learning rates leads to strongly correlated and swapable decoders across spiking LIF and LIF-matched rate networks. A–C** Networks of 2000 neurons were trained on different supervisors over a 40 × 40 grid of points in the $(Q,G)$ parameter space for both spiking LIF and LIF-matched rate networks. The learning rates used were: $\alpha = 5 \times 10^{-6}$, $\alpha = 5 \times 10^{-8}$, and $\alpha = 5 \times 10^{-6}$ for the pitchfork, Ode to Joy, and oscillator respectively. Each set of heatmaps from top to bottom are: the $L_2$ testing error for the spiking

networks, the $L_2$ testing error for the firing rate networks, and the Pearson correlation between the learned decoders of the spiking and rate networks. The stars indicate the most correlated pair of networks with both networks $L_2$ error below a threshold of $5 \times 10^{-6}$, which were used in remaining panels. **D–F** Sample overlaid network outputs (black), sample neuron firing rates (blue), and target supervisor (grey) for both the spiking (solid) and rate (dotted) networks. **G–I** Sample output and neuron dynamics for swapped decoders. The top plots are the output and neuron dynamics for the LIF network with the trained firing rate deocder. The bottom plots are the output and neuron dynamics for the firing rate network with the trained LIF deocder. **J–L** Scatter plot of LIF decoder $\phi_s$ versus firing rate decoder $\phi_R$ with a linear fit.

dynamics for the $(Q,G)$ point with the highest cross-network decoder correlation $\rho_{\phi^R,\phi^S}$, along with the linear regression between $\phi^R$ and $\phi^S$.

When the learning rate was high, we found that decoders could not be fully interchanged between spiking and rate networks (Fig 5G–5I). However, in some cases, the swapped decoders still allowed the networks to reproduce aspects of the target dynamics. Notably, most rate networks using decoders trained in the spiking LIF network performed well (Fig 5E). This one-sided interoperability suggests that the FORCE-learned spiking decoder effectively employs a "noisy" firing rate encoding scheme [4,12]. It also indicates that the decoder learned for the spiking network is more robust, as it generalizes across network types.

## Faster learning improves performance in LIF-matched rate networks but not spiking LIF networks

In the fast learning regime, the divergence of the cross-network decoders suggests that the two networks may be learning different weight structures. To further investigate this, we examined the testing errors and decoder correlations across a grid of $(Q,G)$ values and for different network sizes (Fig 6). We considered the Ode to Joy, Fourier, and Sinusoidal tasks across a $(Q,G)$ grid with four distinct learning rates, ranging from slow ($\alpha = 5 \times 10^{-6}$) to fast ($\alpha = 5 \times 10^0$).

As the learning rate increased, we found that the area of convergence (Fig 6A) and minimal error achieved (Table 2) for the spiking networks were largely unaffected. In contrast, for the LIF-matched rate networks, increasing the learning rate tended to increase the area of convergence (Fig 6A) and decrease the minimal error (Table 3). In the Ode to Joy tasks, higher learning rates ($\alpha > 5 \times 10^{-2}$) resulted in points of high testing error and a decreased area of convergence, suggesting that the FORCE weight updates were unstable [6]. This effect was particularly prominent in higher-dimensional tasks, likely due to higher-dimensional decoder updates and increased feedback from the supervisor, as each readout dimension had its own column in the encoder ($\eta$). As a result, each update step had a greater impact on the network in higher-dimensional supervisors, leading to increased instability in faster learning regimes.

The differences in learning rate dependence between spiking and rate networks suggest that the decoders are optimized differently to leverage their respective neural dynamics. We investigated this further by considering the error scaling as a function of network size for the 5 Hz sinusoidal task at four different learning rates (Fig 6). For a network with $N$ neurons, an efficient spiking network is expected to scale with $N^{-1}$, whereas a firing rate encoding scheme with imprecise spike times would scale with $N^{-1/2}$ [12,30].

To investigate the error scaling further, we trained 21 networks with different random connections for each network type, size, and learning rate (Fig 6B). We then computed the linear line of best fit for the log-log plots to determine the scaling factor $\alpha$ such that the root mean squared error (RMSE) satisfies $RMSE \propto N^\zeta$. For the spiking LIF networks (Fig 6B.I), we found that the learning rate did not significantly affect the scaling factor $\zeta$ or the intercept of the linear fit. Across all learning rates, the scaling factor was approximately –0.5, which, when paired with the spike time variability in the networks (Fig 7), indicates a noisy firing rate encoding

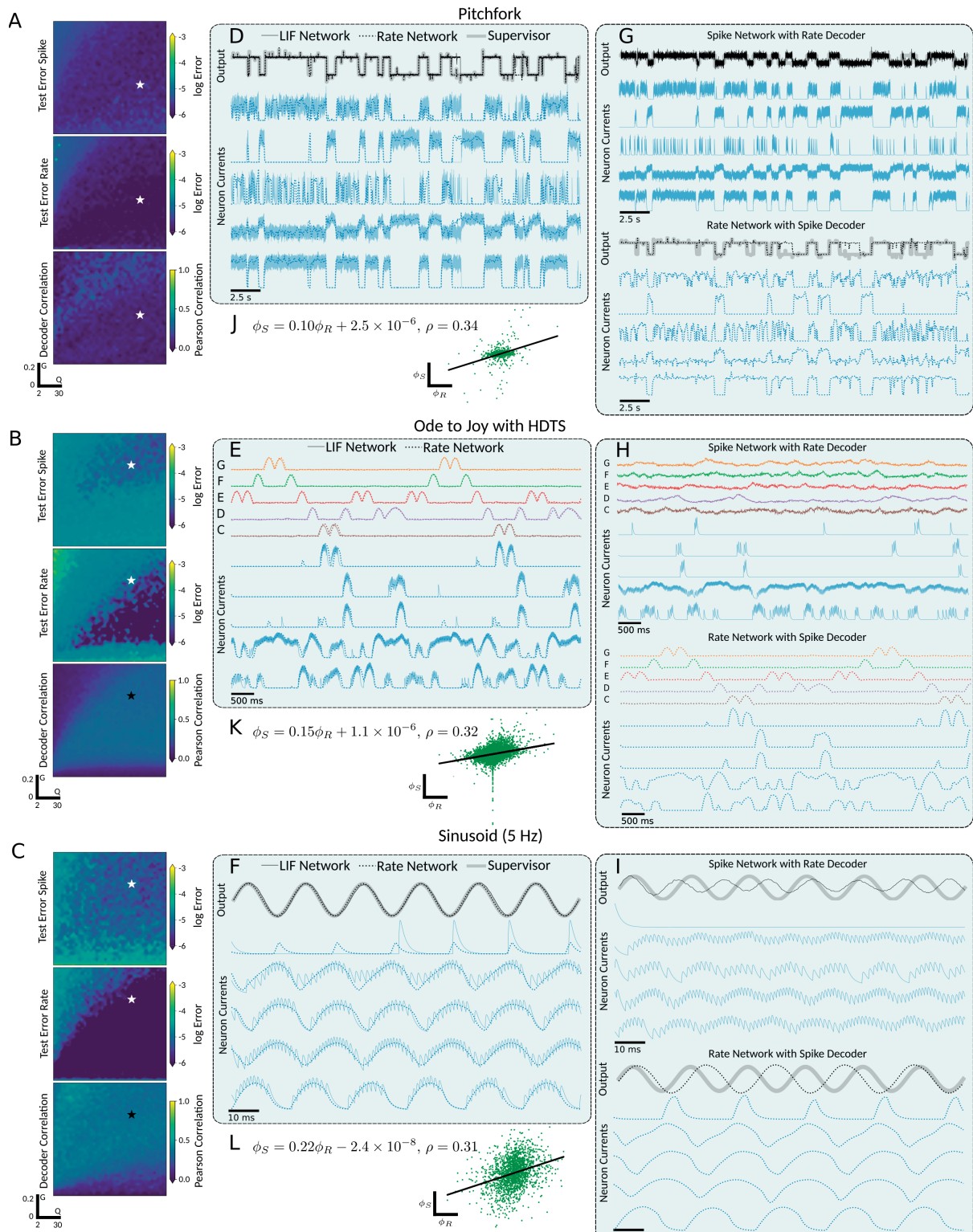

**Fig 5. FORCE training with fast learning rates reduces decoder correlation and swappability. A–C** Row balanced networks of 2000 neurons were train on a supervisors over a $40 \times 40$ grid of points in the $(Q,G)$ parameter space for both LIF and LIF-matched rate networks. The learning rates used were: $\alpha = 5 \times 10^{-4}$, $\alpha = 5 \times 10^{-3}$, and $\alpha = 5$ for the pitchfork, Ode to Joy, and oscillator respectively. Each set of

heatmaps from top to bottom are: the $L_2$ testing error for the LIF networks, the $L_2$ testing error for the rate networks, and the Pearson correlation between the learned decoders of the spiking and rate networks. The stars indicate the most correlated pair of networks with both networks $L_2$ error below a threshold of $5 \times 10^{-6}$, which were used in remaining panels. **D–F** Sample overlaid network outputs (black), sample neuron firing rates (blue), and target supervisor (grey) for both the spiking (solid) and rate (dotted) networks. **G–I** Sample output and neuron dynamics for swapped decoders. The top plots are the output and neuron dynamics for the LIF network with the trained firing rate deocder. The bottom plots are the output and neuron dynamics for the firing rate network with the trained LIF deocder. **J–L** Scatter plot of LIF decoder $\phi_s$ versus firing rate decoder $\phi^R$ with a linear fit.

scheme. In the firing rate network (Fig 6B.II), the scaling factor remained approximately constant across learning rates within the range of $[-0.83, -0.68]$, demonstrating greater coding efficiency than the spiking LIF networks.

Additionally, increasing the learning rate decreased the y-intercept of the log-log linear fit, implying that higher learning rates enhance the reduction in testing error as network size increases. We also computed the decoder correlation across the spiking and rate networks (Fig 6C). As the network size increased, the cross-network decoder correlation converged to a value dependent on the learning rate, with slower learning resulting in higher decoder correlation.

## Spiking network errors stem from variance, rate networks remain stable

The scaling factors found in Fig 6B.I suggest that spiking networks use a firing rate encoding scheme with noisy spike times. This would imply measurable variability in the spike times of the LIF network across multiple repetitions of the same target output signal. Conversely, since the rate model represents the theoretical firing rate, we would expect the neural basis of the firing rate networks to be unaffected by spike-time noise, and consequently, less varied across repeated trials.

To investigate this further, we simulated the networks trained on the 5 Hz sine wave over a $(Q,G)$ grid with different learning rates (Fig 7A) for 100 repetitions (20s) of the supervisor. We then computed the time-averaged bias and variance across the trials. As there are slight differences in the frequency of the target and learned sinusoidal oscillator, there is an apparent phase shift between the network and target outputs, which becomes more pronounced over time (Fig 7A). To account for this, we time-aligned each repetition of the network's output so that the peaks of the sinusoidal oscillator across trials agreed. To compare the variance and bias within each network, we squared the bias, resulting in a decomposition of the mean squared error into the bias squared, the variance, and a cross-term dependent on the probability distribution of the noise [30].

We then computed the proportion of the variance relative to the sum of the variance and bias squared, to determine which dominated the network testing error. In the spiking network, we found that the learning rate had little effect on either bias or variance, neither decreasing the minimal values achieved (Table 4) nor affecting the hyper-parameter space where good values were achieved (Fig 7A). Additionally, in all cases where the networks successfully learned the target dynamics, the bias squared was dominated by the variance (Fig 7A), indicating the primary source of testing error to be variance across trials.

In contrast, the firing rate networks showed a dependence on the learning rate, and the bias squared dominated the variance in all cases where the networks successfully learned the target dynamics (Fig 7A). We found that in the rate networks, faster learning rates led to both the bias and variance achieving good values over a larger area of the $(Q,G)$ hyperparameter space (Fig 7A) and achieving a lower best-case value (Table 4).

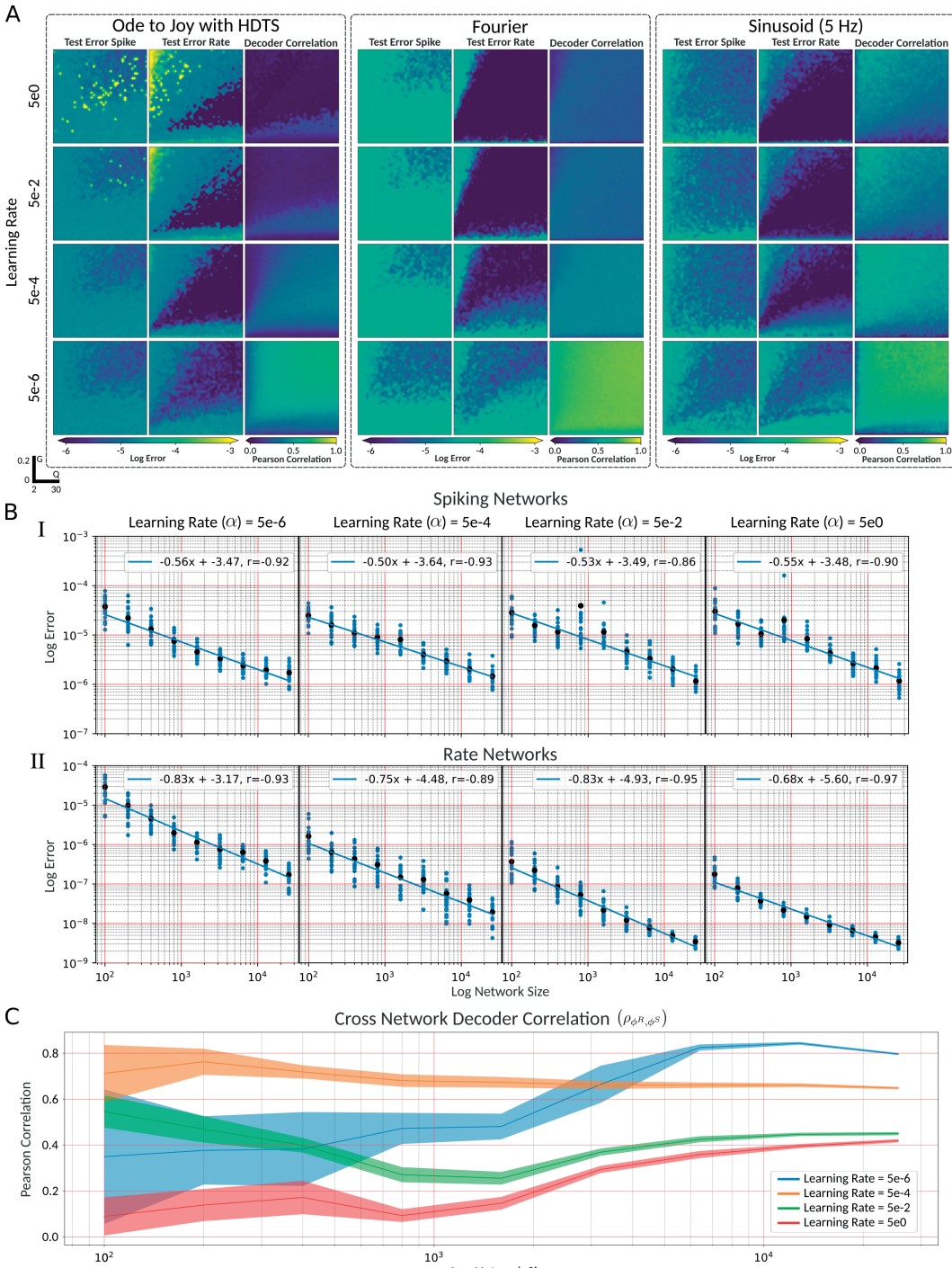

**Fig 6. Fast learning improves performance of LIF-match rate but not spiking LIF networks. A** Networks of 2000 neurons were train on the Ode to Joy, Fourier basis, and sinusoidal tasks over a $40 \times 40$ grid of points in the $(Q,G)$ hyperparameter space with 4 different learning rates for both spiking and rate networks. Within each sub-panel, in order from left to right, we plotted the testing error for the spiking network, the rate network, and the cross network decoder correlation $\rho_{\phi^R, \phi^S}$. **B** For the 5 Hz sinusoidal oscillator task and $(Q,G)$ hyperparameter point (20,0.125), we trained 21 repetitions of randomly initialized networks with sizes in the range $\left[100, 100 \times 2^6\right]$ for 4 different learning rates for both the spiking (**B.I**) and rate model (**B.II**). Each blue point represents a repetition and each black point the mean. The blue lines indicate the linear regression fit with slope and intercepts indicated. **C** Mean Pearson correlation coefficient of decoders across networks $\left(\rho_{\phi^R, \phi^S}\right)$ for simulations in **B**. The shaded area indicates the corrected sampled standard deviation.

**Table 2. Minimal error achieved over (Q,G) grid for LIF network on Fourier, Ode to Joy with HDTS, and 5 Hz sinusoidal oscillator.**

| Learning Rate | $\alpha$ = 5e0 | $\alpha$ = 5e-2 | $\alpha$ = 5e-4 | $\alpha$ = 5e-6 |
|---|---|---|---|---|
| Fourier | 4.5e-06 | 4.9e-06 | 4.0e-06 | 2.9e-06 |
| Ode to Joy with HDTS | 2.4e-06 | 1.7e-06 | 2.1e-06 | 1.9e-06 |
| Simple Oscillator | 1.7e-06 | 1.9e-06 | 1.8e-06 | 2.0e-06 |

**Table 3. Minimal error achieved over (Q,G) grid for LIF-matched rate network on Fourier, Ode to Joy with HDTS, and 5 Hz sinusoidal oscillator.**

| Learning Rate | $\alpha$ = 5e0 | $\alpha$ = 5e-2 | $\alpha$ = 5e-4 | $\alpha$ = 5e-6 |
|---|---|---|---|---|
| Fourier | 8.5e-09 | 2.0e-08 | 1.5e-07 | 1.4e-06 |
| Ode to Joy with HDTS | 3.6e-08 | 3.5e-08 | 5.9e-08 | 4.2e-07 |
| Simple Oscillator | 1.2e-08 | 2.2e-08 | 6.5e-08 | 9.8e-07 |

To understand the difference in variance between the spiking and rate network outputs, we analyzed the neural basis generated by each network for an example network (Fig 7B-C) across repetitions of the target output, both at a fast learning rate ($\alpha = 5 \times 10^0$) and a slow learning rate ($\alpha = 5 \times 10^{-6}$). For the selected networks, we recorded the postsynaptic currents of the neurons in both network types, as well as the spike times for the spiking network. We then time-aligned the postsynaptic currents and spike times for each repetition of the output to create the raster plots in Fig 7B and filtered postsynaptic current plots in Fig 7C.

The spike rasters demonstrated that for both fast and slow learning rates, the inter-trial spike times for each neuron were highly varied but clustered around an average spike time. This variability in spike times resulted in a highly varied filtered postsynaptic current in the spiking networks (Fig 7C), which consequentially led to high variance in the network's output. In contrast, the firing rate networks exhibited highly reproducible postsynaptic currents for both slow and fast learning rates, allowing them to produce network outputs with low cross-trial variance. Averaging the postsynaptic current over trials, we found that spiking and rate networks had similar mean neural dynamics, although the mean spiking postsynaptic currents showed some finer structure due to the clustering of spike times.

In some neurons, the variance around the mean spike times was low enough to prevent significant overlap in subsequent spike times within a repetition of the target signal. This resulted in peaks at the mean firing times and troughs in between. These dynamics are not captured in the instantaneous firing rate model, since the theoretical firing rate depends only on the input current, whereas the peaks in firing result from a correlation in the neuron states across trials. This can be better understood by noting that the firing rate networks use a theoretical instantaneous firing rate while the average postsynaptic current over trials can be understood as the postsynaptic current due to a firing rate average over trials (see [4] or 1.5 of [31] for a discussion of types of firing rate).

Given the difference in variability between the spiking and rate bases, we investigated whether there was a difference in the dynamics of the spiking and rate decoders during the learning phases of FORCE. The FORCE algorithm assumes that the output error is kept small so that the feedback signal is close to the target signal, stabilizing the learning network basis. However, we found that even if the feedback output current to the spiking neural networks is stable, the spiking neural basis has high cross-trial variability. This could lead to instability and convergence issues when using the FORCE algorithm for training a spiking neural network.

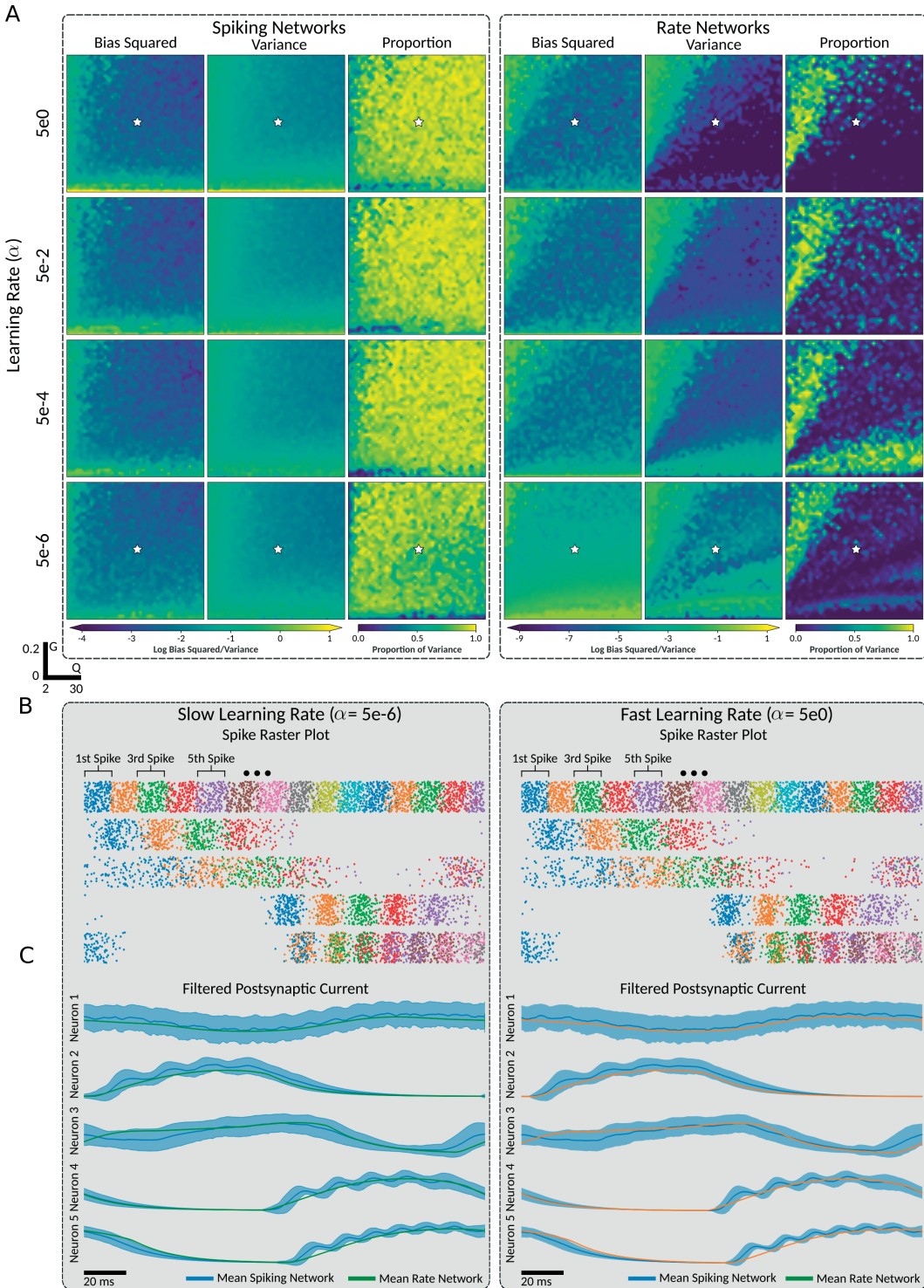

**Fig 7. Variance dominates LIF-spiking networks while bias dominates LIF-matched rate networks in mean squared error decomposition. A** For the 5 Hz sinusoidal oscillator task, we trained networks over a 40 × 40 grid of points in the ($Q$,$G$) hyperparameter space with 4 different learning rates for both LIF and LIF-matched rate networks. For each point in the ($Q$,$G$) space, we simulated the trained network for 100 (20s) repetitions of the sinusoidal output, then computed the cross trail bias and variance of the networks output. Columns of heatmaps within each sub-panel from left to right are: the time averaged bias squared, time averaged variance, and the proportion of the variance to bias squared. The left panel

and right panels contain the plotted values for the spiking network simulations and rate networks, respectively. **B–C** For a selected point (indicated by star in **A**) in the ($Q$,$G$) grid, the corresponding trained network was simulated with both LIF and rate neuron models for slow and fast learning rates for 100 (20s) repetitions of the 5 Hz sinusoidal task. For 5 randomly chosen neurons, the spike times (**B**) for the spiking networks and filtered postsynaptic currents (**C**) for both networks were recorded. To counteract output time-drift, each repetition of the network output was time-aligned to the first peak of the supervisor. **B** Each spike time was represented by a dot, where the colour indicates the order of the spike time within each repetition of the task, indicating their high variability. **C** The postsynaptic filters for both the LIF and rate networks, where the shaded areas indicate the corrected sample standard deviation for both. Note that the deviation for the rate network is also displayed.

**Table 4. Minimal squared bias and variance achieved over ($Q$,$G$) grid for LIF and LIF-matched firing-rate network on 5 Hz sinusoidal oscillator. The minimal variance was only computed over ($Q$,$G$) points that had a corresponding bias squared less than 1e-2.**

| Learning Rate | Minimum Squared Bias | | Minimum Variance | |
|---|---|---|---|---|
| | LIF | Rate | LIF | Rate |
| $\alpha$ = 5e0 | 2.5e-05 | 1.4e-08 | 1.5e-03 | 5.3e-15 |
| $\alpha$ = 5e-2 | 2.8e-05 | 2.3e-08 | 1.5e-03 | 7.1e-11 |
| $\alpha$ = 5e-4 | 2.1e-05 | 1.5e-07 | 1.3e-03 | 2.9e-09 |
| $\alpha$ = 5e-6 | 2.8e-05 | 6.4e-06 | 9.8e-04 | 9.9e-09 |

To investigate this, we recorded a time series for both a spiking and rate network decoders during learning for the 5 Hz sinusoidal signal at a successfully trained point in the ($Q$,$G$) grid (Fig 8). We observed that during training, the firing rate decoders quickly converged as the training error of the network decreased. Conversely, the spiking network decoders failed to stabilize during training and continued to drift randomly even after the network had learned the target dynamics (Fig 8). This decoder instability was accompanied by a more varied testing error which decreased to a point and then continued fluctuating, likely due to the variability of the spiking basis as mentioned. Furthermore, in the fast learning regimes, we often observed that the spiking decoders would destabilize and their size would spike, resulting in better performance with less training (Fig 8).

We also observed that the correlation across the spiking and rate decoders during training initially peaked and then decreased over time (Fig 8). The peak correlation and subsequent decrease were both dependent on the learning rate; slower learning leading to increased peak correlation and a slower decrease. The decrease is likely due to the instability of the spiking decoders, which continue to drift during training.

## Slower learning acts as a low-pass filter of principle components

Recall that the learning rate $\alpha$ is inversely related to the ridge (Tikhonov) regularization parameter $\lambda$. There are two main effects of adjusting $\alpha$ (or equivalently, $\lambda$) on the decoder learned by FORCE: 1) increasing regularization acts as a low-pass filter on the variance of the neural basis, and 2) increasing regularization reduces the magnitude of the decoder vector.

The first effect results in greater similarity between the decoders learned by the spiking and rate networks. This is because the rate network effectively acts as a low-pass filtered version of the spiking network. The second effect is important because reducing decoder magnitude diminishes the impact of trial-to-trial variability in the readout. In FORCE, the readout is fed back into the reservoir, so variance-induced error can destabilize the neural basis. This is especially problematic during training, as the FORCE update rule assumes first-order error dynamics. When this assumption breaks down, decoder updates become less reliable. Both

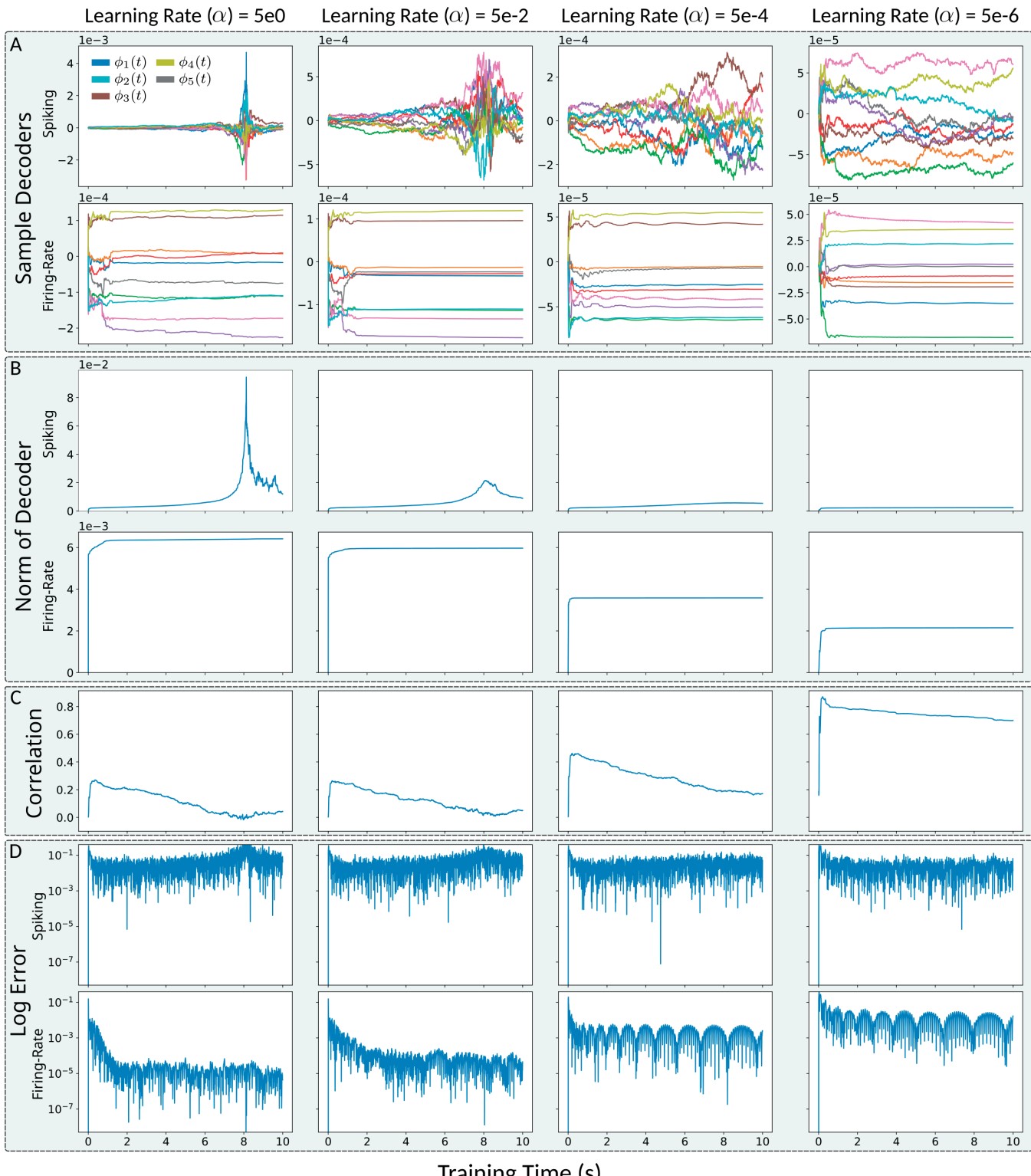

**Fig 8. FORCE trained spiking decoders do not stabilize for fast learning.** Time series data for spiking LIF and LIF-matched rate networks trained on a 5 Hz sinusoidal for 10 s with $(Q, G) = (25, 0.15)$. **A** Sampled decoder elements, **B** Euclidean norm of decoders $\left(\|\phi_{R/S}\|_2\right)$, **C** Pearson correlation across network decoders $(\rho_{\phi^R, \phi^S})$, and **D** log instantaneous error $(\log|x(t) - \hat{x}(t)|)$.

effects together help explain why slower learning rates yield higher correlations between spiking and rate decoders. First, stronger regularization directly aligns the two decoders by filtering out high-frequency (low explanatory power) components. Second, by stabilizing the spiking basis against trial-to-trial fluctuations, slower learning promotes more consistent decoder convergence across network types.

To see the first effect, let $r(t) \in \mathbb{R}^N$ denote the vector of neural basis functions, and let $x(t) \in \mathbb{R}$ be the target signal. The goal of the FORCE algorithm is to approximate $x(t)$ with a readout:

$$\hat{x}(t) = \phi^T r(t). \tag{7}$$

The decoder $\phi$ is learned by minimizing the regularized least-squares loss:

$$L(\phi \mid r, x) = \int_0^T \left( \phi^T r(t) - x(t) \right)^2 \, dt + \lambda \phi^T \phi, \tag{8}$$

over a time interval $[0,T]$. Although FORCE performs online updates that influence the neural basis during learning, we consider here the offline (fixed-basis) solution. The optimal decoder is then:

$$\phi^* = \left( \int_0^T r(t) r^T(t) \, dt + \lambda I \right)^{-1} \left( \int_0^T r(t) x(t) \, dt \right). \tag{9}$$

The matrix inside the inverse is symmetric and positive semi-definite, and can be decomposed as:

$$\int_0^T r(t) r^T(t) \, dt = UDU^T, \tag{10}$$

where $U$ contains the orthonormal eigenvectors and $D$ is a diagonal matrix of the eigenvalues, sorted in descending order. Define $k(t) = U^T r(t)$ as the transformed orthogonal basis. Then the optimal decoder becomes:

$$\phi^* = U \left( D + \lambda I \right)^{-1} \left( \int_0^T k(t) x(t) \, dt \right), \tag{11}$$

and the approximation is given by:

$$\hat{x}(t) = \sum_{n=1}^N \frac{\langle k_n, x \rangle}{D_n + \lambda} k_n(t) = \sum_{n=1}^N \frac{\langle k_n, x \rangle}{\|k_n\|^2 + \lambda} k_n(t), \tag{12}$$

where the inner product is defined by:

$$\langle f, g \rangle = \int_0^T f(t) g(t) \, dt. \tag{13}$$

From this, we see that increasing $\lambda$ (or decreasing $\alpha$) reduces the contribution of components $k_n(t)$ with small $D_n$, effectively suppressing directions of low variance.

To intuitively understand what the basis $k(t)$ represents we consider the discrete case where the basis elements are not functions but vectors of time-samples. Suppose we approximate $r(t)$ using time samples $[r_n(t_1), \ldots, r_n(t_M)]$ and assume zero mean: $\frac{1}{M} \sum_{m=1}^M r_n(t_m) = 0$. Then $[k_n(t_1), \ldots, k_n(t_M)]$ corresponds to the $n$-th principal component of the sample matrix $[r_n(t_m)]_{nm}$ [32]. While this is only an approximation, we can interpret $k_n(t)$ as capturing the $n$-th greatest variance direction in the neural basis $r(t)$.

Given that the spiking and rate bases are empirically highly correlated under stabilizing inputs (S3 Fig), we expect that $k_n(t)$ with large $D_n$ are also more correlated across network types. This is confirmed empirically (S4 Fig). Thus, increasing $\lambda$ reduces the reliance on high-order principal components—those with smaller $D_n$—and promotes consistency across spiking and rate decoders. Given the high variability present in the spiking basis (Fig 7), we also expect that the higher-order components of its orthogonal decomposition primarily reflect noise rather than meaningful signal structure.

To see the second effect, suppose $r(t)$ is partially random, modelled as:

$$r(t) = \bar{r}(t) + \epsilon(t), \tag{14}$$

where $\epsilon(t)$ is a random fluctuation about the mean value $\bar{r}(t)$, described by a family of random variables parameterized in time with mean vector $\mu(t)$ and covariance matrix $\Sigma^2(t)$. The expected readout of the basis is then:

$$\mathbb{E}_\epsilon\left[\hat{x}(t)\right] = \phi^T\left[\bar{r}(t) + \mu(t)\right], \tag{15}$$

and variance:

$$Var_\epsilon\left[\hat{x}(t)\right] = \phi^T\Sigma^2(t)\phi \leq \|\phi\|^2 \left\|\Sigma^2(t)\right\|. \tag{16}$$

This shows that we can reduce the time-dependent variance in the readout by decreasing the magnitude of the decoder. Importantly, increasing regularization (or equivalently, decreasing the learning rate) naturally leads to smaller decoder magnitudes. This has significant implications in the FORCE framework, where the readout is fed back into the reservoir. If this feedback contains excessive variability or error, it can destabilize the basis and degrade network performance. During training, this issue is even more pronounced: since FORCE relies on a first-order approximation of the error for online updates, high variability can lead to inaccurate learning and failure to converge to stable decoder weights.

## Discussion

In this work, we sought to understand the differences between spiking and firing rate based neural networks trained with the FORCE technique. Identifying these differences can help improve spiking neural network learning techniques for engineering tasks (neuromorphic computing) and for modelling learning in biological spiking neural networks. Additionally, understanding what information is preserved or lost when using a rate-based versus a spiking-based model is crucial for developing neural computation theory. Importantly, we suggest that spiking networks trained with the FORCE technique learn to utilize an approximate rate encoding scheme due to the variability in spike times, a result we predict will extend to other least squares based optimization techniques for training spiking neural networks that do not stabilize spike times in some way. Further, we find that rate networks show dramatic improvements when faster learning rates are used, while spiking networks show no improvement, or worse, destabilize.

Additionally, although the dynamics in the spiking and rate networks were always correlated when both networks successfully learned the target dynamics, the correlation between the learned decoder weights depended on the learning rate used in training. We found that when the decoders were correlated (slow learning), we could swap the learned decoders across networks and retain the learned output dynamics. When the decoders were less correlated (fast learning), the firing rate network could still function with the learned spiking

decoder, but the converse was not always true. We also found that the firing rate network was able to achieve lower testing error than the corresponding spiking network. These findings suggest that although both networks performed correlated neural computations when learning was successful, the rate networks achieved more finely tuned solutions that could not work in the spiking networks. Furthermore, since we were able to convert trained spiking networks into corresponding rate networks without loss of performance, the information utilized by the spiking network to perform each task was largely captured by the reduced rate model. This suggests that the spiking networks were using a firing rate based encoding scheme, where the information used for decoding a task was carried by the firing rates.

To further understand the difference in how the spiking and rate networks utilized the neural bases to encode information, we investigated the error scaling as a function of network size. The scaling relationship between error and network size provides insight into how efficiently neural basis elements are being utilized and can suggest the neural encoding scheme. For a perfectly efficient encoding scheme, we would expect the error to scale like $N^{-1}$ where $N$ is the network size [12,30]. Alternatively, in a random noisy spike basis utilizing a rate code, the error would be expected to scale like $N^{-1/2}$ [12,30]. We found an error scaling of about $N^{-1/2}$ in the spiking networks and about $N^{-3/4}$ in the firing rate networks across learning rates. This suggests that the spiking networks are utilizing a noisy rate encoding and demonstrates that the firing rate network is using the neural basis more efficiently. We additionally found that as the learning rate increased, the performance of the spiking networks was largely unaffected, but the firing rate networks were able to achieve lower error at all network sizes. Furthermore, we found that the time-averaged cross-trial variance dominated the time-averaged cross-trial bias squared in the spiking networks. This meant that most of the spiking output error came from the variability in the spiking neural basis. This was corroborated by the high cross-trial time-wise variability found in the spiking basis and spike times. Conversely, in the firing rate networks, the time-averaged cross-trial bias squared dominated the time-averaged cross-trial variance. This indicates that the limiting source of error in the firing rate networks was the expressiveness of the neural basis and decoder. The existence of spike time variability explains the error scaling we previously found [12,33]. This suggests that a scheme closer to a rate encoding, what Denève and Machens called a Poisson rate code, is likely utilized by the spiking networks here.

One important implication of our findings is that, to avoid learning an encoding scheme that approximates the cross-trial averaged firing rate in the LIF network, it is likely necessary to either stabilize spike times relative to the desired learning task or use a learning process that does not rely on recursive least squares, as in the FORCE method. In an unreliable spike basis, the variance will dominate the output error, meaning that the $L^2$ or least squares error will be minimized by focusing on the variance component. In the absence of other factors, such as population-level correlations, this will inherently lead to an encoding scheme that approximates the cross-trial firing rate. Therefore, reducing the variance to allow the bias to dominate the error would enable the emergence of spike encoding strategies that differ from the cross-trial averaged firing rate. We suspect that it is possible to analytically demonstrate that the expected decoder learned by a least-squares-based technique, under certain assumptions about the spike noise, will converge to the optimal decoder for the average cross-trial firing rate basis. However, population-level correlations within the basis must also be taken into account.

We also note that cross-trial variability, like that observed in our study, does not always lead to highly varied network outputs. The issue of variability in the spike trains of FORCE-trained spiking networks has been previously discussed [34]. Another approach to addressing the issue of variability, is to use fast, strong inter-neuron connections that stabilize cross-trial firing rates in neural populations [13–15]. This method relies on neuronal interactions to induce correlations between cross-trial variations in spike timing, which effectively cancel out perturbations to population-level firing rates [34]. As a result, such networks can exhibit highly varied individual neuron spike times without disrupting the overall stability of population-level dynamics, leading to consistent cross-trial network outputs. In the current FORCE framework, all neuronal connections—both inhibitory and excitatory—are determined solely by the fixed random reservoir weights ($G\omega$), fixed random feedback weights ($Q\eta$), and predefined network readout ($\hat{x}(t) \approx x(t)$). However, it may be possible to incorporate or design local learning rules [19,20] to adaptively shape the reservoir and feedback weights, thereby introducing population-level stability within the FORCE framework. There has also been work extending the networks developed by Denève and Machens and using them as the reservoir network in an RLS- or FORCE-based update scheme [35].

There are several possible extensions and notable limitations to this work. The techniques we applied to compare the spiking and corresponding firing rate networks were relatively simple; primarily looking at swap-ability and cross-network correlations. It would be beneficial to use more robust techniques for quantifying the differences in the network types. Due to the increased interest in understanding the largely "black-box" function of ANNs, there have been several attempts to derive techniques to understand them, which could possibly be applied here. For example, work on creating low-dimensional representations of the error landscape for neural networks [36], or on computing the Lyapunov exponents for RNNs [37,38]. Additionally, reservoir techniques have a highly constrained network connectivity and weight structure due to only updating a small subset of the network weights. It would also be interesting to see if our results extend to other spiking neural network training techniques such as full-FORCE [7], E-prop [39], and surrogate gradient descent methods [16,40,41].

We suspect that gradient-based methods may yield different results, as they have access to more complex error targets. However, training corresponding firing-rate networks using these methods presents a challenge, given that the gradient of the firing-rate model diverges (exploding gradient problem) as the firing rate approaches zero. Nevertheless, a viable approach could be to first train a spiking LIF network using a gradient-based method and then convert it into an corresponding firing-rate model. We expect that similar results will be found for full-FORCE due to its similarity to FORCE, despite full-FORCE being designed to create a more stable basis. For example, in full-FORCE based spiking training techniques [42], there was considerable variability within the spike times (raster plots Fig. 4b in [42]). We also expect similar results to hold for FORCE-trained, classification-based tasks, as effective classification requires input currents to drive network activity, much like the tasks examined here. In the work by Nicola and Clopath [18], a high degree of variability was observed in neural voltages for the same inputs—both near and far from the decision boundary. This finding aligns with our observations in this study.

## Materials and methods

### Leaky integrate-and-fire networks

To analyze trained spiking networks with a firing rate match, we used the Leaky-Integrate and Fire (LIF) neuron model with a refractory period:

$$v_i(t) \leftarrow v_{reset}, \qquad \text{if } v_i(t) \geq v_{th}$$

$$\tau_m \dot{v}_i(t) = \begin{cases} 0, & \text{if in a refractory period} \\ -v_i(t) + I_i^S(t), & \text{else} \end{cases} \tag{17}$$

where the sub-index $i$ denotes a specific neuron in the network. The sub-threshold dynamics of the membrane potentials $v_i$ are governed by (17). Once the threshold $v_{th}$ is crossed, the neuron is instantaneously set to a reset voltage $v_{reset}$ and then held at this voltage for a refractory period $\tau_{ref}$. The time that $v_i(t)$ reaches $v_{th}$ corresponds to the firing of a spike in the $i^{th}$ LIF neuron. The parameter $\tau_m$ is the membrane time constant, which dictates the memory of the LIF neuron for subthreshold events. Larger $\tau_m$ lead to longer "storage" and filtering of the sub-threshold dynamics in a neuron. The spiking events are then filtered using the double exponential synaptic filter:

$$\dot{h}_i^S = -\frac{h_i^S}{\tau_r} + \frac{1}{\tau_r \tau_d} \sum_{t_{ik} < t} \delta(t - t_{ik}) \tag{18}$$

$$\dot{r}_i^S = -\frac{r_i^S}{\tau_d} + h_i^S \tag{19}$$

where $\delta(\cdot)$ is the Dirac delta function. The parameter $\tau_r$ is the synaptic rise time and $\tau_d$ the synaptic decay time, which control how quickly the postsynaptic current pulse rises and decays. The neuron parameters are listed in Table 1. The current of the $i^{th}$ neuron, $I_i^S(t)$ is the sum of a background bias current $I_{bias}$, which we took to be the rheobase ($I_{bias} = -40$ mV), and the synaptic input currents:

$$I_i^S(t) = I_{bias} + \sum_{j=1}^{N} \omega_{ij}^S r_j^S(t) \tag{20}$$

where $\omega^S \in \mathbb{R}^{N \times N}$ is the synaptic connectivity matrix. Note that a unit resistance is absorbed into the input currents. The synaptic connectivity matrix is composed of a static portion $G\omega^0$ and a learned portion $Q\eta \left[\phi^S\right]^T$ as:

$$\omega^S = G\omega^0 + Q\eta \left[\phi^S\right]^T. \tag{21}$$

The static portion of the matrix was scaled by a recurrent connectivity strength parameter $G$ scaling a sparse random matrix $\omega^0 \in \mathbb{R}^{N \times N}$. For our investigation we generated $\omega^0$ from a normal distribution with mean 0 and variance of $\left(Np^2\right)^{-1}$ where $p$ is the sparsity of the matrix and $N$ the number of neurons in the network. This scales the weights to be in proportion to the inverse of the square-root of the number of connections [18]. To introduce sparsity into the reservoir connections, each weight $\omega_{ij}^0$ was randomly set to be zero with probability $1-p$. Further, the sample mean of each row in the matrix was explicitly set to be 0, which encourages homogeneity in the neural firing rates [18]. This is done by first generating a random matrix $\tilde{\omega}^0$ from the normal distribution $\mathcal{N}(0, p(N)^{-1/2})$ with the desired level of sparsity then

defining:

$$\omega_{ij}^0 = \tilde{\omega}_{ij}^0 - \frac{1}{N}\sum_{j=1}^{N}\tilde{\omega}_{ij}^0. \tag{22}$$

The learned portion of the connectivity matrix consisted of a constant scalar strength parameter $Q$, a static encoder $\eta \in \mathbb{R}^{N \times M}$, and a learned decoder $\phi^S \in \mathbb{R}^{N \times M}$, where $M$ is the dimension of the network output. By introducing the $Q$ and $G$ parameters we can control the macro-scale dynamics of the network; either inducing more (less) chaos by increasing (decreasing) $G$ or creating more (less) structured heterogeneity in the neural firing rates by increasing (decreasing) $Q$ [22,23]. The components of the encoder are drawn from a uniform distribution over the $M$ dimensional unit square $[-1,1]^M$. The learned linear decoder $\phi^S$ serves the primary task of decoding the neural dynamics to produce the network output:

$$\hat{x}^S(t) = \sum_{i=1}^{N}\phi_i^S r_i^S \tag{23}$$

but also works in conjunction with the encoder to stabilize the network dynamics via the feedback connections.

## Firing rate networks

If the current $I_i$ is constant or very slowly varying relative to $v_i(t)$, in (17), the solution to the differential equation for the LIF neuron is computed as:

$$v_i(t) = (v_i(0) - I_i)\exp\left[\frac{t}{\tau_m}\right] - I_i \tag{24}$$

for some initial voltage $v(0)$. Then, the time to the first spike $t_{spike}$, can be computed as the time required to reach the threshold voltage $v_{th}$ from the reset voltage $v_{reset}$ given by:

$$t_{spike} = -\tau_m \ln\left[\frac{I_i - v_{th}}{I_i - v_{reset}}\right]. \tag{25}$$

This value plus the refractory period $\tau_{ref}$ is then the time between spikes given a constant current. As the spike rate is the inverse of the time between spikes, the firing rate for a LIF neuron with an input current $I_i$ is given by:

$$R(I_i) = \left[\tau_{ref} - \tau_m \ln\left(\frac{I_i - v_{th}}{I_i - v_{reset}}\right)\right]^{-1} \tag{26}$$

where $R(I_i)$ is often called the transfer function (or $f$–$I$ curve) of neuron $i$. The firing rate, $R(I_i)$, is then used to approximate the filtered spike train as [1]:

$$\dot{h}_i^R = -\frac{h_i^R}{\tau_r} + \frac{1}{\tau_r \tau_d}R_i(I_i)$$
$$\dot{r}_i^R = -\frac{r_i^R}{\tau_d} + h_i^R$$

which will be valid under the assumption that the synapses are much slower than the spiking dynamics or that the specific spike-times are not important for the network dynamics [1].

Thus, for a network of $N$ leaky-integrate-and-fire neurons, the firing rate system is determined by:

$$I_i^R(t) = \sum_{j=1}^{N} \omega_{ij}^R r_j^R(t) + I_{bias} \tag{27}$$

$$R(t) = \left[ \tau_{ref} - \tau_m \ln \left( \frac{I_i^R(t) - v_{th}}{I_i^R(t) - v_{reset}} \right) \right]^{-1} \tag{28}$$

$$\dot{h}_i^R = -\frac{h_i^R}{\tau_r} + \frac{1}{\tau_r \tau_d} R(t) \tag{29}$$

$$\dot{r}_i^R = -\frac{r_i^R}{\tau_d} + h_i^R. \tag{30}$$

The firing rate connectivity matrix $\omega^R \in \mathbb{R}^{N \times N}$ is defined analogously to that of the spiking connectivity matrix $\omega^S$ with the only difference being the introduction of the rate network decoder $\phi^R$. When all the parameters in Eqs (27)–(30) are matched to their spiking network counterpart, the system (27)–(30) predicts the dynamics of the firing rates of the spiking network.

## FORCE training

The primary objective of FORCE training is to approximate some target dynamical system $x(t)$ with the network output $\hat{x}(t)$, using a sample of the dynamics of $x(t)$ over an interval. To do this, FORCE uses the Recursive Least Squares (RLS) technique to minimize the squared error loss function:

$$L(\hat{x}, x; t) = \sum_{j=1}^{M} \int_0^t \left( \hat{x}_j(s) - x_j(s) \right)^2 ds + \lambda \phi^T \phi \tag{31}$$

where $\lambda$ is a scalar regularization parameter that penalizes large values of the decoders. The RLS algorithm works by dynamically reducing the error in the network output and target dynamics:

$$e(t) = \hat{x}(t) - x(t), \tag{32}$$

at a series of time steps by performing discrete updates to the decoder $\phi$. The update at time step $t$ with step size $\Delta t$ is computed according to:

$$\phi(t) = \phi(t - \Delta t) - e^T(t) P(t) r(t)$$

$$P(t) = P(t - \Delta t) - \frac{P(t - \Delta t) r(t) r^T(t) P(t - \Delta t)}{1 + r^T(t) P(t - \Delta t) r(t)}$$

where $P(t)$ is the discrete approximation to the inverse cross-correlation of the neural basis, plus regularization matrix:

$$P(t)^{-1} \approx \int_0^t r(s) r(s)^T ds + \lambda I_N. \tag{33}$$

The decoder is initialized as $\phi(0) = 0$ and the correlation matrix with:

$$P(0) = \frac{I_N}{\lambda} = \alpha I_N. \tag{34}$$

where we can view the parameter $\alpha$ as the learning rate for the FORCE algorithm [6]. We note that the regularization parameter in the least squares formalization $\lambda$ is the inverse of the learning rate $\alpha$ in the RLS FORCE framework. We make this distinction since, in the least squares framework, the optimization occurs over a fixed time interval so $\lambda$ adjusts the magnitude of $\phi$ but in the RLS framework, $\alpha$ influences how large updates to $\phi$ are at each time step. As we are dealing mainly with RLS, we will mainly refer to the learning rate $\alpha$ but will occasionally refer to the regularization $\lambda$ as they serve a related function.

## Fig 3 methods

Networks of spiking neurons and their parameter-matched rate networks were separately trained using the FORCE technique to produce various dynamical systems. For all simulations, an integration time step of 0.05ms was used in a forward Euler integration scheme. The network size $N$, graph sparsity $p$, reservoir strength $G$, feedback strength $Q$, training time $T_{train}$, testing time $T_{test}$, and RLS update interval $\Delta t$ are as listed in Table 5.

The sinusoidal supervisor depicted in Fig 3 is a 5 Hz sinusoidal oscillator with unit amplitude. To illustrate the spontaneous activity before training, the network ran for 0.6s before RLS training commenced. Then, RLS was activated for 1s, followed by deactivation for 0.6s to test the network. The complex oscillator (Fig 3C) results from the product of 1 Hz and 2 Hz sinusoidal oscillators. For the Ode to Joy supervisor, a 5-dimensional signal representing the G, F, E, D, and C notes of the first bar of the song was used. Each pulse of the supervisor was generated by the positive portion of a 1 Hz or 2 Hz sinusoidal waveform, representing a half or quarter note in the song based on the pulse duration, respectively. During training, the signal was repeated 20 times, totalling 80 s of simulated training time, and then tested for an additional 2 repetitions of the signal (8 s), with the last repetition (4 s) utilized in Fig 3D–3E.

The pitchfork supervisor was generated from the pitchfork normal form

$$\gamma \dot{x} = rx - x^3 + p(t)$$

for some perturbation function in time $p(t)$. The parameter $\gamma$ is a time constant that adjusts how quickly the system responds to perturbations due to $p(t)$ and the positive parameter $r$ controls the location of the stable fixed points in the unperturbed system ($p(t) \equiv 0$), which occur at $\pm\sqrt{r}$. The parameters $r$ and $\gamma$ were fixed at $r = 1$ and $\gamma = 0.01$. To generate the perturbations, $p(t)$, a random sequence of square pulses was created where the time between pulses was uniformly sampled from [0.1,0.5] while the length of the pulse was uniformly sampled from [0.005,0.01], and the height of the pulses were sampled from a normal distribution with standard deviation 2 and zero mean.

The Lorenz system was generated from the following equations:

$$\dot{x} = \sigma (y - x) \tag{35}$$

**Table 5. Parameters for Fig 3.**

| Supervisor | $N$ | $p$ | $G$ | $Q$ | $T_{train}$ | $T_{test}$ | $\Delta t$ | $\tau_d$ |
|---|---|---|---|---|---|---|---|---|
| Simple Oscillator | 2000 | 0.4 | 0.19 | 25 | 1 s | 0.6 s | 0.25 ms | 20 ms |
| pitchfork | 2000 | 0.4 | 0.16 | 28 | 120 s | 28 s | 0.5 ms | 20 ms |
| Complex Oscillator | 2000 | 0.3 | 0.16 | 28 | 5 s | 3 s | 0.5 ms | 20 ms |
| Ode to Joy | 2000 | 0.4 | 0.16 | 28 | 80 s | 4 s | 0.5 ms | 20 ms |
| Lorenz System | 5000 | 0.3 | 0.1 | 20 | 200 s | 200 s | 0.25 ms | 100 ms |

$$\dot{y} = x\left(\rho - z\right) - y \tag{36}$$

$$\dot{z} = xy - \beta z \tag{37}$$

with parameters $\rho = 28$, $\sigma = 5$, and $\beta = \frac{7}{3}$. Along with the Lorenz-like trajectories shown in Fig 3F, we computed the return maps for both the trained spiking and rate networks. The Lorenz maps were computed by first smoothing the raw network output $\hat{z}$ by convolution with a window filter of length 21 time steps, then computing the local maximums $\hat{z}_n$ of the resulting signal using the SciPy argrelextrema [43] function with order 10000. The maps are then formed by plotting $\hat{z}_{n+1}$ vs $\hat{z}_n$.

## Figs 4 and 5 methods

The pitchfork, Ode to Joy, and sinusoidal supervisors were produced in the same way as those in Fig 3, with the addition that the Ode to Joy supervisors also included a high-dimensional temporal signal (HDTS) as used by Nicola and Clopath 2017 [18]. The HDTS consists of a series of extra components added to the supervisor to increase its dimensionality, that acts as a clock. These components help stabilize the network by driving subsets of the network into assemblies of synchronized activity, which helps disambiguate repeating note sub-sequences in the Ode to Joy song [18].

To produce an $m$-dimensional HDTS, we divided the time interval $[0, T]$ into $m$ subintervals $I_n = \left[T\left(\frac{n-1}{m}\right), T\left(\frac{n}{m}\right)\right]$ for $n = 1, 2, \ldots, m$. The $n$th component of the HDTS contained a pulse in the interval $I_n$. Here, we used the positive portion of a sine wave, yielding the additional components:

$$x_n(t) = \begin{cases} \left|\sin\left(\frac{m\pi t}{T}\right)\right| & t \in I_n \\ 0 & \text{otherwise.} \end{cases} \tag{38}$$

For the Ode to Joy examples, an HDTS with 16 components was used.

The testing error of a decoded output $\hat{\boldsymbol{x}}(t)$ for each point in the $(Q, G)$ grid for an $M$ dimensional supervisor $x(t)$ on the interval $[0, T]$ was computed by the $L^2$ norm:

$$E\left(\hat{\boldsymbol{x}}(t), \boldsymbol{x}(t)\right) = \frac{1}{M} \sum_{i=1}^{M} \sqrt{\int_0^T \left(\hat{x}_i(t) - x_i(t)\right)^2 \, dt} \tag{39}$$

The Pearson correlation coefficient between the $M$ dimensional spiking decoder $\phi^S$ and the firing rate decoder $\phi^R$ for a network of size $N$ at each point in the $(Q, G)$ grid was computed as:

$$\rho_{\phi^R, \phi^S} = \frac{\sum_{i=1}^{M} \sum_{j=1}^{N} \left(\phi_{ij}^R - \overline{\phi^R}\right)\left(\phi_{ij}^S - \overline{\phi^S}\right)}{\sqrt{\sum_{i=1}^{M} \sum_{j=1}^{N} \left(\phi_{ij}^R - \overline{\phi^R}\right)^2} \sqrt{\sum_{i=1}^{M} \sum_{j=1}^{N} \left(\phi_{ij}^S - \overline{\phi^S}\right)^2}} \tag{40}$$

where $\overline{\phi}$ is the mean of $\phi$ over all components and dimensions. For each supervisors, we then chose the $(Q, G)$ point with the highest $\rho_{\phi^R, \phi^S}$ where the testing errors $E(\hat{x})$ for both the spiking and rate networks was less than $5 \times 10^{-5}$. For the resulting point $(Q^*, G^*)$, the spiking network was simulated with the firing rate decoder $\phi^R$ and the firing rate network was simulated with the spiking decoder $\phi^S$. The network size $N$, sparsity $p$, training time $T_{train}$, testing time $T_{test}$, and RLS update interval $\Delta t$ for each supervisor are as listed in Table 6.

## Fig 6 methods

The Ode to Joy with HDTS and sinusoidal supervisors were constructed as previously described. The Fourier supervisors consisted of 9 sinusoidal waves, with frequencies evenly distributed from 0.5 to 4.5, each serving as its own component of the supervisors. The overall Fourier supervisor is described by:

$$x_n(t) = \sin(n\pi t) \tag{41}$$

where $x_n$ is the $n^{th}$ component of the supervisor and $n = 1, 2, \ldots, 9$.

To compute the error scaling and cross network decoder correlations, 21 different spiking and rate networks were trained on the 5 Hz sinusoidal supervisors for network sizes ranging from $N = 100$ to $N = 25600$. Each network was row balanced with reservoir and feedback parameters $(Q, G) = (20, 0.125)$, trained with $T_{train} = 5$ s and tested with $T_{test} = 0.4$ s. The testing error and cross networks decoder correlation was calculated as previously described.

## Fig 7 methods

For each point on the $(Q,G)$ grid, a spiking and rate network with 2000 neurons was trained on a 5 Hz sinusoidal supervisor for 10 (2s) repetitions and then tested for 100 (20s) repetitions. The 20s interval was segmented into 100 evenly spaced 0.2s intervals, each containing a single repetition of the sinusoidal supervisor. Each repetition was time-aligned by computing the maximal value in the sub-interval and rolling the output signal so that the maximal value occurred at the expected time of $t = 0.05$s. For the spiking network, the signal was first smoothed by convolving it with a square window function of width 0.0250s before computing the maximal value and then time-aligning the original unsmoothed signal. Let $\hat{x}_i(t)$ be the $i^{th}$ time aligned repetition and $n_{rep}$ the number of repetitions, the bias squared was then computed by:

$$\mathrm{Bias}\,(\hat{x})^2 = \frac{1}{T}\int_0^T \left(\frac{1}{n_{rep}}\sum_{i=1}^{n_{rep}}\hat{x}_i(t) - x(t)\right)^2 dt \tag{42}$$

the variance is computed as:

$$\mathrm{Var}\,(\hat{x}) = \frac{1}{T}\sum_{j=1}^{n_{rep}}\int_0^T \left(\hat{x}_j(t) - \frac{1}{n_{rep}}\sum_{i=1}^{n_{rep}}\hat{x}_i(t)\right)^2 dt \tag{43}$$

and the proportion of variance as:

$$\frac{\mathrm{Var}\,(\hat{x})}{\mathrm{Bias}\,(\hat{x})^2 + \mathrm{Var}\,(\hat{x})}. \tag{44}$$

**Table 6. Training parameters for Figs 4 and 5.**

| Supervisor | N | p | $T_{train}$ (s) | $T_{test}$ (s) | $\Delta t$ (ms) |
|---|---|---|---|---|---|
| pitchfork | 2000 | 0.4 | 120 | 120 | 5 |
| Ode to Joy with HDTS | 2000 | 0.4 | 80 | 8 | 5 |
| Simple Oscillator | 2000 | 0.4 | 2 | 2 | 5 |

## Supporting information

**S1 Fig. Chaotic regime for FORCE-trained LIF and LIF-matched rate networks. A-B** Neural currents for networks of 200 LIF neurons and their corresponding LIF-matched rate neurons, demonstrated chaotic behaviour for reservoir strength parameter $G$>0. **C** Phase portrait of the first two neurons in the rate network simulated for 50 s, displaying chaotic behaviour for $G$>0. **D** Networks of 2000 LIF and LIF-matched rate neurons were trained over a 40 × 40($Q, G$) parameter grid with $G \in [0, 2]$ and $Q \in [2, 100]$. For sufficiently large $G$, neither network was able to learn.
(TIFF)

**S2 Fig. FORCE-trained networks struggle with fast supervisors and short training durations. A** Networks of 2000 LIF and LIF-matched rate neurons were trained to generate sine waves of increasing frequency. As the frequency increased, both networks exhibited reduced ability to learn the supervisor. **B** Networks of 2000 LIF and LIF-matched rate neurons were trained to generate a 5 Hz sine wave with varying training durations. When the training period was very short (containing only a single cycle of the supervisor), neither network was able to learn.
(TIFF)

**S3 Fig. LIF and LIF-matched rate networks exhibit correlated neural bases when driven but are chaotic without input.** Networks of 2000 LIF and LIF-matched rate neurons were simulated for 1s across different reservoir strengths ($G$) and feedback strengths ($Q$), both with and without driving input. In the absence of input, both networks exhibited chaotic dynamics and low cross-network correlations. When driven, the cross-network readout and neural bases became correlated. **A–B** Sample readouts and neural basis elements from both networks. **C–F** Cross-network correlations of sampled neural bases and readouts.
(TIFF)

**S4 Fig. LIF and LIF-matched rate networks have highly correlated low order principle components.** Networks of 2000 LIF and LIF-matched rate networks were trained with FORCE on the 5 Hz sinusoidal supervisor for 4s with learning rate $\alpha$ =5e-6, reservoir strengths $G$ = 0.1, and feedback strengths $Q$ = 15, for 10 different seed values. We then plot the averaged: absolute correlation in the orthogonal basis elements, LIF eigenvalue, and Rate eigenvalue. The shaded regions represent the standard deviation. The lower order orthogonal basis elements have much higher associated eigenvalues and so explain much more of the variability in the original basis. The early basis elements are also highly correlated across networks types.
(TIFF)

**S1 Text. LIF and LIF-matched rate networks exhibit correlated neural bases when driven but are chaotic without input.** Methods for S3 Fig.
(PDF)

## Author contributions

**Conceptualization:** Thomas Robert Newton, Wilten Nicola.

**Formal analysis:** Thomas Robert Newton.

**Funding acquisition:** Wilten Nicola.

**Investigation:** Thomas Robert Newton.

**Software:** Thomas Robert Newton.

**Supervision:** Wilten Nicola.

**Visualization:** Thomas Robert Newton.

**Writing – original draft:** Thomas Robert Newton.

**Writing – review & editing:** Thomas Robert Newton, Wilten Nicola.

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
