## [Decision Letter · Decision Letter 0]

12 Nov 2024

PCOMPBIOL-D-24-01502FORCE trained spiking networks do not benefit from faster learning while parameter matched rate networks doPLOS Computational Biology

Dear Dr. Nicola,

Thank you for submitting your manuscript to PLOS Computational Biology. After careful consideration, we feel that it has merit but does not fully meet PLOS Computational Biology's publication criteria as it currently stands. Therefore, we invite you to submit a revised version of the manuscript that addresses the points raised during the review process.

The referees raised serious issues concerning, particularly:

- the very restricted part of the state-space of LIF network considered

- Details of the match between the rate and spiking networks

- task dependencies

Please validate the results in requested simulations and provide a thorough response to the reviewers comments.

Please submit your revised manuscript within 60 days Jan 12 2025 11:59PM. If you will need more time than this to complete your revisions, please reply to this message or contact the journal office at ploscompbiol@plos.org. Please include the following items when submitting your revised manuscript:

We look forward to receiving your revised manuscript.

Kind regards,

Anna LevinaAcademic EditorPLOS Computational BiologyHugues BerrySection EditorPLOS Computational Biology

Feilim Mac Gabhann

Editor-in-Chief

PLOS Computational Biology Jason Papin

Editor-in-Chief

PLOS Computational Biology

**Journal Requirements:** **Additional Editor Comments (if provided):****Reviewers' comments:** Reviewer's Responses to Questions

**Comments to the Authors:**

Reviewer #1: Summary:

The study compares two types of networks trained under FORCE learning: a spiking LIF network and a rate-based version of the former. The rate-based approximation relies on a classical derivation: with constant current, the inter spike interval (ISI) of deterministic LIF is constant. Both networks are trained on the same toy tasks: Lorenz attractor, bit-flip or reproducing period inputs. The two networks are matching very strongly in the specific regime considered. The most prominent difference highlighted is that rate-based networks learn better under high learning rates.

General comment:

The match between spiking and non-spiking networks here is somewhat remarkable given that it is uneasy to find an accurate one-to-one mapping between spiking networks are rates. However the derivation relies on the classical derivation of stationary ISI for the case of a single neuron and this interesting extension to the network dynamics is easily explainable since the current varies slowly. I believe this is engineered in the simulations of this paper and would not be true for most spiking models (see point A). Also the scope of spiking networks studying here is restricted and unlikely to other more useful applications of spiking network models (see point B).

Detailed comments:

A) The classical results on stationary spike-intervals in deterministic LIF networks is sufficient here by design for two reasons: (1) the target signals are relatively slow to match the quasi-stationary current and ISI hypothesis, (2) the network are set in a very sub-chaotic state (G~0.1) probably near the mean-field limit where weights are weak and firing is driven by collective activity and reducing the transient effects of isolated spikes (see [1] for a theoretical mapping of stochastic spiking networks and rate based networks in the mean field regime). The point (2) is probably even consolidated by FORCE learning which projects back a low-dimensional input using low-dimensional factors.

So one essential problem here is that (2) is not commented on or highlighted, but I strongly suspect this is an important assumption here. An easy check would be to study higher ranges for G which are closer to the chaotic regime (G~1 is the parametrization is nice, otherwise showing that dynamics start to vary in a chaotic fashion at some point). The whole point of the paper is strongly defeated if the distances between spiking and rate based networks decreases although both networks solve the tasks with near-chaotic G which is the most standard choice.

B) I find the scope of analysis quite limited in the paper. Only the case of FORCE learning is studied under tasks which are relatively very simple in comparison with contemporary application of spiking networks. For instance the Od to the Joy task requires repeating (hence "overfitting") a periodic signal and does not require generalization. This optimization is far more trivial in comparison with the fitting to neural recordings [2] or machine learning applications as done with surrogate gradient techniques). Typically it is likely that the quasi-stationary current assumption would not hold in these harder cases.

C) The title suggests that the strongest result is that spiking networks do not benefit from fast learning rate. I think this is however mildly surprising, and if this is such a strong result then it could deserve more analysis. I suggest two possible hypotheses: the very fast learning rates in Figure 8 are sufficient to "learn" with hundreds of milliseconds. I wonder whether this alone might not induce fast dynamic changes which are interfering with the quasi-stationary ISI assumptions. Second hypothesis, the spiking network behaves as a "noisy" rate base network so more repetition of the periods are needed to average learning signals.

[1] Mesoscopic population equations for spiking neural networks with synaptic short-term plasticity

Schmutz et al. 2020

https://link.springer.com/article/10.1186/s13408-020-00082-z

[2] Trial matching: capturing variability with data-constrained spiking neural networks

Sourmpis et. al 2023

https://openreview.net/forum?id=LAbxkhkjbD¬eId=P91RSu6siy

Reviewer #2: Please see the attachment for my concerns and input.

Reviewer #3: Overall, I value the paper and I believe it addresses a relevant question, that many of us that have worked (or tried to) with spiking FORCE networks will recognise. I appreciate the methods explanation per figure, which makes it easy for the reader to reproduce the results.

I have two main issues:

Some of the structure is not quite clear.

Parts of the system are explained in the results, with a more extensive explanation in the Methods. However, no references are made between the two, which makes it difficult for the reader to understand the ‘Network Models’ part of the results. For instance, the difference between tau_r and tau_ref is not explained, it is not explained that the spikes are in equation 3 (basically, how to go from v to r), and what is tau_d? For eta and phi the dimensions are given, but it is not explained what they are, and are the brackets in equation 6 a function (i.e. eta is a function of phi) or are they just there for the transpose? What is P (the update role is given, but that it is related to the rate correlation matrix is stated nowhere). In the end, most of this can be deduced from the methods, but it would be nice if it were either explained more explicitly, or contained references to where it is explained. In particular, the learning rate alpha is not in any of the equations, which is very confusing, as this is one of the key points of the paper.

The tasks (pitchfork, ode to joy, etc) are also not explained in the results, which makes this hard to follow.

A more fundamental issue is that of task dependence. Figure 6 shows clear task dependence: increasing the learning rate alpha (so going from bottom to top) increases the the area of convergenc, but not for Ode to Joy. So it is not task independent. Yet, in figure 6, 7 and 8 you only go on with the 5 Hz oscillations (which is quite slow for a neural system). The scaling with network size is very interesting, and I can understand that you cannot test this for all G,S values and tasks, but why was this particular parameter set and task chosen? Do you expect your results to change with different parameter values or with a different task, as we say that the area of convergence is task dependent. Moreover, next to different dynamical systems, Nicola and Clopath show that their system can also function for different tasks, for instance as a classifier. Maybe the authors could discuss whether their results would also hold for classifier tasks? Finally, trial to trial variability is not always the same as jitter, as in the predictive coding networks of Machens and Denève that the authors cite demonstrate: they have a large trial to trial variability without a loss of information, whereas jitter would increase the error a lot. So I suspect that the results might not always hold over tasks, and that spiking network might perform better for tasks where faster timescales are required. As I understand that it is not feasible to simulate too many tasks and parameters, this should at least be discussed.

Minor points:

I think the term ‘LIF-rate’ network is confusing: the F in LIF stands for fire, and that is exactly what rate neurons do not do. Could you maybe use a term like ‘LIF-matched rate network’ or something?

In the introduction, there are some local learning rules in predictive coding networks that I believe should be mentioned, as it shows that the representations can be learned by local Hebbian rules:

Bourdoukan, R., Barrett, D. G. T., Machens, C. K., & Denève, S. (2012). Learning optimal spike-based representations. In P. Bartlett, F. C. N. Pereira, C. J. C. Burges, L. Bottou, & K. Q. Weinberger (Eds.), Advances in Neural Information Processing Systems 25 (pp. 2294–2302).

Brendel, W., Bourdoukan, R., Vertechi, P., Machens, C. K., & Denève, S. (2020). Learning to represent signals spike by spike. PLoS Computational Biology, 16(3), Article 3. https://doi.org/10.1371/journal.pcbi.1007692

At the end of the results section, it is not completely clear what you mean here by a basis. Could that be specified more explicitly?

Figure 4: the fact that the stars are at the edge of your figures, suggests that if you would have chosen a larger range of G and Q values, there might have been even more correlated values. Does the correlation go back up again, or does it keep decreasing? Why was this range of G and Q chosen?

Fig 4 and 5: why were these alpha values chosen, what was it baed on? And from the methods I read that a HDTS signal was used to stabilise the network. This is not trivial to me: could this be explained in the results or discussion? What was is based on, why does that work, why is it needed?

In figure 7 there is an orange line that is not mentioned in the caption, whereas there is mention of a green line that is not in the figure

Figure 8a needs a legend (what are the colours?).

I add the pdf with some highlighted typos (as there are no line numbers), and I few sentences I could not follow.

**Have the authors made all data and (if applicable) computational code underlying the findings in their manuscript fully available?**

Reviewer #1: Yes

Reviewer #2: Yes

Reviewer #3: None

PLOS authors have the option to publish the peer review history of their article (what does this mean?). If published, this will include your full peer review and any attached files.

Reviewer #1: No

Reviewer #2: **Yes: **Nasir Ahmad

Reviewer #3: **Yes: **Fleur Zeldenrust

 **Figure resubmission:**While revising your submission, please upload your figure files to the Preflight Analysis and Conversion Engine (PACE) digital diagnostic tool, https://pacev2.apexcovantage.com/. PACE helps ensure that figures meet PLOS requirements. To use PACE, you must first register as a user. Registration is free. Then, login and navigate to the UPLOAD tab, where you will find detailed instructions on how to use the tool. If you encounter any issues or have any questions when using PACE, please email PLOS at figures@plos.org. Please note that Supporting Information files do not need this step. If there are other versions of figure files still present in your submission file inventory at resubmission, please replace them with the PACE-processed versions.   
---

## [Decision Letter · Decision Letter 1]

24 Apr 2025

PCOMPBIOL-D-24-01502R1

FORCE trained spiking networks do not benefit from faster

learning while parameter matched rate networks do

PLOS Computational Biology

Dear Dr. Nicola,

Thank you for submitting your manuscript to PLOS Computational Biology. After careful consideration, we feel that it has merit but does not fully meet PLOS Computational Biology's publication criteria as it currently stands. Therefore, we invite you to submit a revised version of the manuscript that addresses the points raised during the review process.

Please submit your revised manuscript within 30 days Jun 24 2025 11:59PM. If you will need more time than this to complete your revisions, please reply to this message or contact the journal office at ploscompbiol@plos.org. Please include the following items when submitting your revised manuscript:

We look forward to receiving your revised manuscript.

Kind regards,

Anna Levina

Academic Editor

PLOS Computational Biology

Hugues Berry

Section Editor

PLOS Computational Biology

**Additional Editor Comments :**

The revised manuscript has addressed the majority of the referees' concerns. However, both reviewers raise a remaining issue regarding the precise formulation and limits of the equivalence between spiking and rate models. In light of their comments, I believe the manuscript is close to publication and could be accepted after minor revision. The authors are encouraged to further clarify their terminology around model equivalence and, if feasible, include additional discussion or exploratory results related to the regimes where this equivalence may fail.

**Journal Requirements:**

1) We have noticed that you have uploaded Supporting Information files, but you have not included a list of legends. Please add a full list of legends for your Supporting Information files after the references list.

2) Please amend your detailed Financial Disclosure statement. This is published with the article. It must therefore be completed in full sentences and contain the exact wording you wish to be published.

3) Please provide a completed 'Competing Interests' statement, including any COIs declared by your co-authors. If you have no competing interests to declare, please state "The authors have declared that no competing interests exist". Otherwise please declare all competing interests beginning with the statement "I have read the journal's policy and the authors of this manuscript have the following competing interests:"

**Reviewers' comments:**

Reviewer's Responses to Questions

Reviewer #1: I believe that the paper is in an acceptable stage for publication.

I would like to thank the authors for taking my comments into considerations, although my original review was harsh and badly written. I am sorry for that.

I am still uneasy with the usage of the word "equivalent" in Introduction and Results. Probably for similar reasons as reviewer 2. If used, this term should be given a contextual definition that is mathematically clear and unambiguous, because a naive reader could understand that there exist some kind of general and standard equivalence between LIF and rate. This is not true in general. As explained in Material and Methods, the "equivalence" that is used here comes mathematically from the derivation using stationary inputs. This should be spelled out early in the Results, I am not really satisfied with the current sentences line 93 to 97 which are mathematically vague and imprecise. Hence I suggest the following change:

Suggestion: line 95 one could add a sentence in the spirit of: "[... range of tasks [6, 18].] This equivalence is derived in Material and Methods using the steady-state firing rate of an isolated LIF neuron with constant input current, and remarkably the resulting pair of LIF and firing rate networks share intimate properties when trained with FORCE. [By using exactly parameter matched networks,...].

It is up to the authors to apply this correction, or another one but I believe that some clarification early in results is necessary.

Reviewer #2: I thank the authors for their engagement with the reviews and applaud the beneficial changes made to the paper. It is definitely more clear all around.

I believe this paper now has sufficient clarity for a reader to grasp the main points and is on an interesting and important topic. However, I am still hesitant to support immediate publication because I believe that the investigation is left short of completeness. In the rebuttal to my review, the authors point out that:

“the firing rate approximation (interchanging rates with filtered spikes) is more generally true for slowly varying currents through dynamical systems averaging, or when the network filters (e.g. synaptic filters) are sufficiently long”

and later that

“We leave an analytical derivation of this point, which relates the weight magnitude to the stability of the spiking versus the rate network as a future point of work.”

I believe that these two comments (and the origin of my questions which prompted them) point to the only hole left in this work. Specifically, that it appears there are specific regimes of parameter space (not just learning rates) in which the equivalence of the rate networks breaks down and that large learning rates push the networks to these regimes. Expanding on this would allow a much stronger take-away from such a work beyond advising smaller learning rates and would elevate its contribution significantly.

This could be achieved by looking into whether in the breakdown cases, the requirement for equivalence (slowly varying currents) is broken and then attempting to define (analytically or empirically) at what limit of weight-magnitude and/or spiking-rate, this breakdown occurs. This could provide a much deeper insight and takeaway from this work. As a casual (throw-away) example of how such a take-away could look, it could indeed be that weight magnitudes are the determining factor - if so, regularisation of weight magnitudes might ensure network equivalence and/or that spike rate must also be regularized. Any such outcome would turn this work into a seriously depthful contribution.

Given that I don't see that there are any problems with this work, but instead that there is greater potential here, I have opted to submit a recommendation of 'minor revision' so that the editor and authors can consider whether this is sufficiently important for them. I would be okay with an outright acceptance at this stage but, as I mentioned, would be even happier to see this work reach its potential for impact.

**Have the authors made all data and (if applicable) computational code underlying the findings in their manuscript fully available?**

Reviewer #1: Yes

Reviewer #2: Yes

PLOS authors have the option to publish the peer review history of their article (what does this mean?). If published, this will include your full peer review and any attached files.

Reviewer #1: No

Reviewer #2: **Yes: **Nasir Ahmad

**Figure resubmission:**
---

## [Editor Report · Decision Letter 2]

10 Jun 2025

Dear Dr. Nicola,

We are pleased to inform you that your manuscript 'Comparison of FORCE trained spiking and rate neural networks shows spiking networks learn slowly with noisy, cross-trial firing rates' has been provisionally accepted for publication in PLOS Computational Biology.

Best regards,

Anna Levina

Academic Editor

PLOS Computational Biology

Hugues Berry

Section Editor

PLOS Computational Biology

---

## [Editor Report · Acceptance letter]

PCOMPBIOL-D-24-01502R2

Comparison of FORCE trained spiking and rate neural networks shows spiking networks learn slowly with noisy, cross-trial firing rates

Dear Dr Nicola,

I am pleased to inform you that your manuscript has been formally accepted for publication in PLOS Computational Biology. Your manuscript is now with our production department and you will be notified of the publication date in due course.

With kind regards,

Zsofia Freund
